# FULLY HYPERBOLIC CONVOLUTIONAL NEURAL NETWORKS FOR COMPUTER VISION

**Ahmad Bdeir**[1,*]**, Kristian Schwethelm**[2,*,†] **& Niels Landwehr**[1]
[1] Data Science Department, University of Hildesheim
[2] Chair for Artificial Intelligence in Medicine, Technical University of Munich
`{bdeira, schwethelm, landwehr}@uni-hildesheim.de`

## ABSTRACT

Real-world visual data exhibit intrinsic hierarchical structures that can be represented effectively in hyperbolic spaces. Hyperbolic neural networks (HNNs) are a promising approach for learning feature representations in such spaces. However, current HNNs in computer vision rely on Euclidean backbones and only project features to the hyperbolic space in the task heads, limiting their ability to fully leverage the benefits of hyperbolic geometry. To address this, we present HCNN, a fully hyperbolic convolutional neural network (CNN) designed for computer vision tasks. Based on the Lorentz model, we generalize fundamental components of CNNs and propose novel formulations of the convolutional layer, batch normalization, and multinomial logistic regression. Experiments on standard vision tasks demonstrate the promising performance of our HCNN framework in both hybrid and fully hyperbolic settings. Overall, we believe our contributions provide a foundation for developing more powerful HNNs that can better represent complex structures found in image data. Our code is publicly available at https://github.com/kschwethelm/HyperbolicCV.

## 1 INTRODUCTION

Representation learning is a fundamental aspect of deep neural networks, as obtaining an optimal representation of the input data is crucial. While Euclidean geometry has been the traditional choice for representing data due to its intuitive properties, recent research has highlighted the advantages of using hyperbolic geometry as a geometric prior for the feature space of neural networks. Given the exponentially increasing distance to the origin, hyperbolic spaces can be thought of as continuous versions of trees that naturally model tree-like structures, like hierarchies or taxonomies, without spatial distortion and information loss (Nickel & Kiela, 2018; Sarkar, 2012). This is compelling since hierarchies are ubiquitous in knowledge representation (Noy & Hafner, 1997), and even the natural spatial representations in the human brain exhibit a hyperbolic geometry (Zhang et al., 2023).

Leveraging this better representative capacity, hyperbolic neural networks (HNNs) have demonstrated increased performance over Euclidean models in many natural language processing (NLP) and graph embedding tasks (Peng et al., 2022). However, hierarchical structures have also been shown to exist in images. Mathematically, Khrulkov et al. (2020) have found high $\delta$-hyperbolicity in the final embeddings of image datasets, where the hyperbolicity quantifies the degree of inherent tree-structure. Extending their measurement to the whole model reveals high hyperbolicity in intermediate embeddings as well (see Appendix D.1). Intuitively, hierarchies that emerge within and across images can be demonstrated on the level of object localization and object class relationships. A straightforward example of the latter is animal classification hierarchy, where species is the lowest tier, preceded by genus, family, order, etc. Similarly, on a localization level, humans are one example: the nose, eyes, and mouth are positioned on the face, which is a part of the head, and, ultimately, a part of the body. This tree-like localization forms the basis of part-whole relationships and is strongly believed to be how we parse visual scenes (Biederman, 1987; Hinton, 1979; Kahneman et al., 1992).

In light of these findings, recent works have begun integrating hyperbolic geometry into vision architectures (Mettes et al., 2023; Fang et al., 2023). Specifically, they rely on the Poincaré ball

---

*Equal contribution. †Work done while at University of Hildesheim.

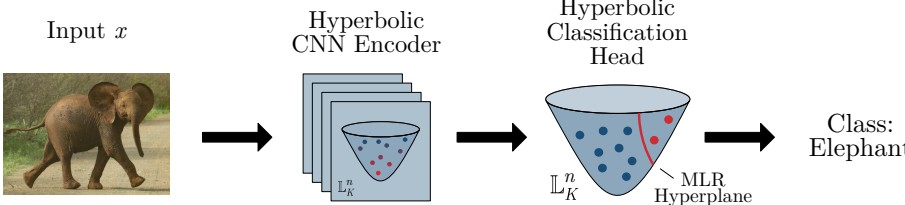

Figure 1: In contrast to hybrid HNNs that use a Euclidean CNN for feature extraction, our HCNN learns features in hyperbolic spaces in every layer, fully leveraging the benefits of hyperbolic geometry. This leads to better image representations and performance.

and the Lorentz model as descriptors of hyperbolic space and formalize hyperbolic translations of neural network layers. This is challenging due to ill-defined hyperbolic analogs of, e.g., addition, multiplication, and statistical measures. Currently, most HNN components are only available in the Poincaré ball as it supports the gyrovector space with basic vector operations. However, due to its hard numerical constraint, the Poincaré ball is more susceptible to numerical instability than the Lorentz model (Mishne et al., 2022), which motivates introducing the missing layers for the Lorentz model. Moreover, HNNs in computer vision have been limited to hybrid architectures that might not fully leverage the advantages of hyperbolic geometry as they rely on Euclidean encoders to learn hyperbolic representations. Until now, hyperbolic encoder architectures are missing in computer vision, although prevalent in NLP and graph applications (Peng et al., 2022).

In this work, we present HCNN, a fully hyperbolic framework for vision tasks that can be used to design hyperbolic encoder models. We generalize the ubiquitous convolutional neural network (CNN) architecture to the Lorentz model, extend hyperbolic convolutional layers to 2D, and present novel hyperbolic formulations of batch normalization and multinomial logistic regression. Our methodology is general, and we show that our components can be easily integrated into existing architectures. Our contributions then become three-fold:

1. We propose hybrid (HECNN) and fully hyperbolic (HCNN) convolutional neural network encoders for image data, introducing the fully hyperbolic setting in computer vision.

2. We provide missing Lorentzian formulations of the 2D convolutional layer, batch normalization, and multinomial logistic regression.

3. We empirically demonstrate the performance potential of deeper hyperbolic integrations in experiments on standard vision tasks, including image classification and generation.

## 2 RELATED WORK

**Hyperbolic image embeddings** Previous research on HNNs in computer vision has mainly focused on combining Euclidean encoders and hyperbolic embeddings. This approach involves projecting Euclidean embeddings onto the hyperbolic space in the task heads and designing task-related objective functions based on hyperbolic geometry. Such simple hybrid architectures have been proven effective in various vision tasks like recognition (Yu et al., 2022; Khrulkov et al., 2020; Liu et al., 2020; Guo et al., 2022), segmentation (Hsu et al., 2020; Atigh et al., 2022), reconstruction/generation (Mathieu et al., 2019; Nagano et al., 2019; Ovinnikov, 2019; Qu & Zou, 2022), and metric learning (Ermolov et al., 2022; Yan et al., 2021; Yue et al., 2023). However, there remains the discussion of whether the single application of hyperbolic geometry in the decoder can fully leverage the present hierarchical information. In contrast, HE/HCNN also learns latent hyperbolic feature representations in the encoder, potentially magnifying these benefits. We also forgo the typically used Poincaré ball in favor of the Lorentz model, as it offers better stability and optimization (Mishne et al., 2022). For a complete overview of vision HNNs and motivations, refer to (Mettes et al., 2023; Fang et al., 2023).

**Fully hyperbolic neural networks** Designing fully hyperbolic neural networks requires generalizing Euclidean network components to hyperbolic geometry. Notably, Ganea et al. (2018) and Shimizu et al. (2020) utilized the Poincaré ball and the gyrovector space to generalize various layers, including fully-connected, convolutional, and attention layers, as well as operations like split, concatenation,

and multinomial logistic regression (MLR). Researchers have also designed components in the Lorentz model (Nickel & Kiela, 2018; Fan et al., 2022; Chen et al., 2021; Qu & Zou, 2022), but crucial components for vision, like the standard convolutional layer and the MLR classifier, are still missing. Among the hyperbolic layer definitions, fully hyperbolic neural networks have been built for various tasks in NLP and graph applications (Peng et al., 2022). However, no hyperbolic encoder architecture has yet been utilized in computer vision. Our work provides formulations for missing components in the Lorentz model, allowing for hyperbolic CNN vision encoders. Concurrently, van Spengler et al. (2023) proposed a fully hyperbolic Poincaré CNN.

**Normalization in HNNs** There are few attempts at translating standard normalization layers to the hyperbolic setting. To the best of our knowledge, there is only a single viable normalization layer for HNNs, i.e., the general Riemannian batch normalization (Lou et al., 2020). However, this method is not ideal due to the slow iterative computation of the Fréchet mean and the arbitrary re-scaling operation that is not based on hyperbolic geometry. The concurrent work on Poincaré CNN (van Spengler et al., 2023) only solved the first issue by using the Poincaré midpoint. In contrast, we propose an efficient batch normalization algorithm founded in the Lorentz model, which utilizes the Lorentzian centroid (Law et al., 2019) and a mathematically motivated re-scaling operation.

**Numerical stability of HNNs** The exponential growth of the Lorentz model's volume with respect to the radius can introduce numerical instability and rounding errors in floating-point arithmetic. This requires many works to rely on 64-bit precision at the cost of higher memory and runtime requirements. To mitigate this, researchers have introduced feature clipping and Euclidean reparameterizations (Mishne et al., 2022; Guo et al., 2022; Mathieu et al., 2019). We adopt these approaches to run under 32-bit floating point arithmetic and reduce computational cost.

## 3 BACKGROUND

This section summarizes the mathematical background of hyperbolic geometry (Cannon et al., 2006; Ratcliffe, 2006). The $n$-dimensional hyperbolic space $\mathbb{H}_K^n$ is a Riemannian manifold $(\mathcal{M}^n, \mathfrak{g}_x^K)$ with constant negative curvature $K < 0$, where $\mathcal{M}^n$ and $\mathfrak{g}_x^K$ represent the manifold and the Riemannian metric, respectively. There are isometrically equivalent models of hyperbolic geometry. We employ the Lorentz model because of its numerical stability and its simple exponential/logarithmic maps and distance functions. Additionally, we use the Poincaré ball for baseline implementations. Both hyperbolic models provide closed-form formulae for manifold operations, including distance measures, exponential/logarithmic maps, and parallel transportation. They are detailed in Appendix A.

**Lorentz model** The $n$-dimensional Lorentz model $\mathbb{L}_K^n = (\mathcal{L}^n, \mathfrak{g}_x^K)$ models hyperbolic geometry on the upper sheet of a two-sheeted hyperboloid $\mathcal{L}^n$, with origin $\overline{\mathbf{0}} = [\sqrt{-1/K}, 0, \cdots, 0]^T$ and embedded in $(n+1)$-dimensional Minkowski space (see Figure 2). Based on the Riemannian metric $\mathfrak{g}_x^K = \mathrm{diag}(-1, 1, \ldots, 1)$, the manifold is defined as

$$\mathcal{L}^n := \{\boldsymbol{x} \in \mathbb{R}^{n+1} \mid \langle \boldsymbol{x}, \boldsymbol{x} \rangle_\mathcal{L} = \frac{1}{K}, \ x_t > 0\}, \quad (1)$$

with the Lorentzian inner product

$$\langle \boldsymbol{x}, \boldsymbol{y} \rangle_\mathcal{L} := -x_t y_t + \boldsymbol{x}_s^T \boldsymbol{y}_s = \boldsymbol{x}^T \mathrm{diag}(-1, 1, \cdots, 1)\boldsymbol{y}. \quad (2)$$

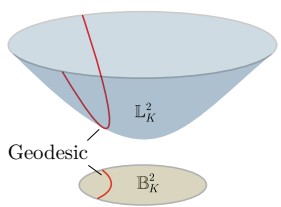

Figure 2: Comparison of Lorentz and Poincaré model.

When describing points in the Lorentz model, we inherit the terminology of special relativity and call the first dimension the *time component* $x_t$ and the remaining dimensions the *space component* $\boldsymbol{x}_s$, such that $\boldsymbol{x} \in \mathbb{L}_K^n = [x_t, \boldsymbol{x_s}]^T$ and $x_t = \sqrt{||\boldsymbol{x_s}||^2 - 1/K}$.

## 4 FULLY HYPERBOLIC CNN (HCNN)

We aim to give way to building vision models that can fully leverage the advantages of hyperbolic geometry by learning features in hyperbolic spaces. For this, we generalize Euclidean CNN components

to the Lorentz model, yielding one-to-one replacements that can be integrated into existing architectures. In the following, we first define the cornerstone of HCNNs, i.e., the Lorentz convolutional layer, including its transposed variant. Then, we introduce the Lorentz batch normalization algorithm and the MLR classifier. Finally, we generalize the residual connection and non-linear activation.

## 4.1 LORENTZ CONVOLUTIONAL LAYER

**Hyperbolic feature maps** The convolutional layer applies vector operations to an input feature map containing the activations of the previous layer. In Euclidean space, arbitrary numerical values can be combined to form a vector. However, in the Lorentz model, not all possible value combinations represent a point that can be processed with hyperbolic operations ($\mathbb{L}_K^n \subset \mathbb{R}^{n+1}$).

We propose using channel-last feature map representations throughout HCNNs and adding the Lorentz model's time component as an additional channel dimension. This defines a hyperbolic feature map as an ordered set of $n$-dimensional hyperbolic vectors, where every spatial position contains a vector that can be combined with its neighbors. Additionally, it offers a nice interpretation where an image is an ordered set of color vectors, each describing a pixel.

**Formalization of the convolutional layer** We define the convolutional layer as a matrix multiplication between a linearized kernel and a concatenation of the values in its receptive field, following Shimizu et al. (2020). Then, we generalize this definition by replacing the Euclidean operators with their hyperbolic counterparts in the Lorentz model.

Given a hyperbolic input feature map $\mathbf{x} = \{\boldsymbol{x}_{h,w} \in \mathbb{L}_K^n\}_{h,w=1}^{H,W}$ as an ordered set of $n$-dimensional hyperbolic feature vectors, each describing image pixels, the features within the receptive field of the kernel $\mathbf{K} \in \mathbb{R}^{m \times n \times \tilde{H} \times \tilde{W}}$ are $\{\boldsymbol{x}_{h'+\delta\tilde{h},w'+\delta\tilde{w}} \in \mathbb{L}_K^n\}_{\tilde{h},\tilde{w}=1}^{\tilde{H},\tilde{W}}$, where $(h', w')$ denotes the starting position and $\delta$ is the stride parameter. Now, we define the Lorentz convolutional layer as

$$\boldsymbol{y}_{h,w} = \text{LFC}(\text{HCat}(\{\boldsymbol{x}_{h'+\delta\tilde{h},w'+\delta\tilde{w}} \in \mathbb{L}_K^n\}_{\tilde{h},\tilde{w}=1}^{\tilde{H},\tilde{W}})), \tag{3}$$

where HCat denotes the concatenation of hyperbolic vectors, and LFC denotes a Lorentz fully-connected layer performing the affine transformation and parameterizing the kernel and bias, respectively (see Appendix A). Additionally, we implement padding using origin vectors, the analog of zero vectors in hyperbolic space. The LFC layer is similar to Chen et al. (2021) but does not use normalization as it is done through the hyperbolic batch normalization formulated below.

**Extension to the transposed setting** The transposed convolutional layer is usually used in encoder-decoder architectures for up-sampling. A convolutional layer carries out a transposed convolution when the correct local connectivity is established by inserting zeros at certain positions. Specifically, when stride $s > 1$, then $s - 1$ zero vectors are inserted between the features. We refer to Dumoulin & Visin (2016) for illustrations. Under this relationship, the Lorentz transposed convolutional layer is a Lorentz convolutional layer with changed connectivity through origin padding.

## 4.2 LORENTZ BATCH NORMALIZATION

Given a batch $\mathcal{B}$ of $m$ features $\boldsymbol{x}_i$, the traditional batch normalization algorithm (Ioffe & Szegedy, 2015) calculates the mean $\boldsymbol{\mu}_\mathcal{B} = \frac{1}{m}\sum_{i=1}^m \boldsymbol{x}_i$ and variance $\boldsymbol{\sigma}_\mathcal{B}^2 = \frac{1}{m}\sum_{i=1}^m (\boldsymbol{x}_i - \boldsymbol{\mu}_\mathcal{B})^2$ across the batch dimension. Then, the features are *re-scaled* and *re-centered* using a parameterized variance $\boldsymbol{\gamma}$ and mean $\boldsymbol{\beta}$ as follows

$$\text{BN}(\boldsymbol{x}_i) = \boldsymbol{\gamma} \odot \frac{\boldsymbol{x}_i - \boldsymbol{\mu}_\mathcal{B}}{\sqrt{\boldsymbol{\sigma}_\mathcal{B}^2 + \epsilon}} + \boldsymbol{\beta}. \tag{4}$$

At test time, running estimates approximate the batch statistics. They are calculated iteratively during training: $\boldsymbol{\mu}_t = (1 - \eta)\boldsymbol{\mu}_{t-1} + \eta\boldsymbol{\mu}_\mathcal{B}$ and $\boldsymbol{\sigma}_t^2 = (1 - \eta)\boldsymbol{\sigma}_{t-1}^2 + \eta\boldsymbol{\sigma}_\mathcal{B}^2$, with $\eta$ and $t$ denoting momentum and the current iteration, respectively. We generalize batch normalization to the Lorentz model using the Lorentzian centroid and the parallel transport operation for re-centering, and the Fréchet variance and straight geodesics at the origin's tangent space for re-scaling.

**Re-centering**  To re-center hyperbolic features, it is necessary to compute a notion of mean. Usually, the Fréchet mean is used (Lou et al., 2020), which minimizes the expected squared distance between a set of points in a metric space (Pennec, 2006). Generally, the Fréchet mean must be solved iteratively, massively slowing down training. To this end, we propose to use the centroid with respect to the squared Lorentzian distance, which can be calculated efficiently in closed form (Law et al., 2019). The weighted Lorentzian centroid, which solves $\min_{\boldsymbol{\mu} \in \mathbb{L}_K^n} \sum_{i=1}^m \nu_i d_{\mathcal{L}}^2(\boldsymbol{x}_i, \boldsymbol{\mu})$, with $\boldsymbol{x}_i \in \mathbb{L}_K^n$ and $\nu_i \geq 0, \sum_{i=1}^m \nu_i > 0$, is given by

$$\boldsymbol{\mu} = \frac{\sum_{i=1}^m \nu_i \boldsymbol{x}_i}{\sqrt{-K} \left| \|\sum_{i=1}^m \nu_i \boldsymbol{x}_i\|_{\mathcal{L}} \right|}. \tag{5}$$

In batch normalization, the mean is not weighted, which gives $\nu_i = \frac{1}{m}$. Now, we shift the features from the batch's mean $\boldsymbol{\mu}_{\mathcal{B}}$ to the parameterized mean $\boldsymbol{\beta}$ using the parallel transport operation $\mathrm{PT}_{\boldsymbol{\mu}_{\mathcal{B}} \to \boldsymbol{\beta}}^K(\boldsymbol{x})$. Parallel transport does not change the variance, as it is defined to preserve the distance between all points. Finally, the running estimate is updated iteratively using the weighted centroid with $\nu_1 = (1 - \eta)$ and $\nu_2 = \eta$.

**Re-scaling**  For re-scaling, we rely on the Fréchet variance $\sigma^2 \in \mathbb{R}^+$, defined as the expected squared Lorentzian distance between a point $\boldsymbol{x}_i$ and the mean $\boldsymbol{\mu}$, and given by $\sigma^2 = \frac{1}{m} \sum_{i=1}^m d_{\mathcal{L}}^2(\boldsymbol{x}_i, \boldsymbol{\mu})$ (Kobler et al., 2022). In order to re-scale the batch, features must be moved along the geodesics connecting them to their centroid, which is generally infeasible to compute. However, geodesics intersecting the origin are very simple, as they can be represented by straight lines in tangent space $\mathcal{T}_{\bar{\boldsymbol{0}}} \mathbb{L}_K^n$. This is reflected by the equality between the distance of a point to the origin and the length of its corresponding tangent vector ($d_{\mathcal{L}}(\boldsymbol{x}, \bar{\boldsymbol{0}}) = \|\log_{\bar{\boldsymbol{0}}}^K(\boldsymbol{x})\|$). Using this property, we propose to re-scale features by first parallel transporting them towards the origin $\mathrm{PT}_{\boldsymbol{\mu}_{\mathcal{B}} \to \bar{\boldsymbol{0}}}^K\left(\log_{\boldsymbol{\mu}_{\mathcal{B}}}^K(\boldsymbol{x})\right)$, making the origin the new centroid and straightening the relevant geodesics. Then, a simple multiplication re-scales the features in tangent space. Finally, parallel transporting to $\boldsymbol{\beta} \in \mathbb{L}_K^n$ completes the algorithm and yields the normalized features. The final algorithm is formalized as

$$\mathrm{LBN}(\boldsymbol{x}) = \exp_{\boldsymbol{\beta}}^K\left(\mathrm{PT}_{\bar{\boldsymbol{0}} \to \boldsymbol{\beta}}^K\left(\gamma \cdot \frac{\mathrm{PT}_{\boldsymbol{\mu}_{\mathcal{B}} \to \bar{\boldsymbol{0}}}^K\left(\log_{\boldsymbol{\mu}_{\mathcal{B}}}^K(\boldsymbol{x})\right)}{\sqrt{\sigma_{\mathcal{B}}^2 + \epsilon}}\right)\right). \tag{6}$$

### 4.3 Lorentz MLR classifier

In this section, we consider the problem of classifying instances that are represented in the Lorentz model. A standard method for multi-class classification is multinomial logistic regression (MLR). Inspired by the generalization of MLR to the Poincaré ball (Ganea et al., 2018; Shimizu et al., 2020) based on the distance to margin hyperplanes, we derive a formulation in the Lorentz model.

**Hyperplane in the Lorentz model**  Analogous to Euclidean space, hyperbolic hyperplanes split the manifold into two half-spaces, which can then be used to separate instances into classes. The hyperplane in the Lorentz model is defined by a geodesic that results from the intersection of an $n$-dimensional hyperplane with the hyperboloid in the ambient space $\mathbb{R}^{n+1}$ (Cho et al., 2019). Specifically, for $\boldsymbol{p} \in \mathbb{L}_K^n$ and $\boldsymbol{w} \in \mathcal{T}_{\boldsymbol{p}} \mathbb{L}_K^n$, the hyperplane passing through $\boldsymbol{p}$ and perpendicular to $\boldsymbol{w}$ is given by

$$H_{\boldsymbol{w}, \boldsymbol{p}} = \{\boldsymbol{x} \in \mathbb{L}_K^n \mid \langle \boldsymbol{w}, \boldsymbol{x} \rangle_{\mathcal{L}} = 0\}. \tag{7}$$

This formulation comes with the non-convex optimization condition $\langle \boldsymbol{w}, \boldsymbol{w} \rangle_{\mathcal{L}} > 0$, which is undesirable in machine learning. To eliminate this condition, we use the Euclidean reparameterization of Mishne et al. (2022), which we extend to include the curvature parameter $K$ in Appendix B.1. In short, $\boldsymbol{w}$ is parameterized by a vector $\bar{\boldsymbol{z}} \in \mathcal{T}_{\bar{\boldsymbol{0}}} \mathbb{L}_K^n = [0, a\boldsymbol{z}/\|\boldsymbol{z}\|]$, where $a \in \mathbb{R}$ and $\boldsymbol{z} \in \mathbb{R}^n$. As $\boldsymbol{w} \in \mathcal{T}_{\boldsymbol{p}} \mathbb{L}_K^n$, $\bar{\boldsymbol{z}}$ is parallel transported to $\boldsymbol{p}$, which gives

$$\boldsymbol{w} := \mathrm{PT}_{\boldsymbol{0} \to \boldsymbol{p}}^K (\overline{\boldsymbol{z}}) = [\sinh(\sqrt{-K}a)||\boldsymbol{z}||, \cosh(\sqrt{-K}a)\boldsymbol{z}]. \tag{8}$$

Inserting Eq. 8 into Eq. 7, the formula of the Lorentz hyperplane becomes

$$\tilde{H}_{\boldsymbol{z},a} = \{\boldsymbol{x} \in \mathbb{L}_K^n \mid \cosh(\sqrt{-K}a)\langle \boldsymbol{z}, \boldsymbol{x}_s \rangle - \sinh(\sqrt{-K}a)\,||\boldsymbol{z}||\,x_t = 0\}, \tag{9}$$

where $a$ and $\boldsymbol{z}$ represent the distance and orientation to the origin, respectively.

Finally, we need the distance to the hyperplane to quantify the model's confidence. It is formulated by the following theorem, proven in Appendix B.2.

**Theorem 1** *Given $a \in \mathbb{R}$ and $\boldsymbol{z} \in \mathbb{R}^n$, the minimum hyperbolic distance from a point $\boldsymbol{x} \in \mathbb{L}_K^n$ to the hyperplane $\tilde{H}_{\boldsymbol{z},a}$ defined in Eq. 9 is given by*

$$d_{\mathcal{L}}(\boldsymbol{x}, \tilde{H}_{\boldsymbol{z},a}) = \frac{1}{\sqrt{-K}} \left| \sinh^{-1} \left( \sqrt{-K} \frac{\cosh(\sqrt{-K}a)\langle \boldsymbol{z}, \boldsymbol{x}_s \rangle - \sinh(\sqrt{-K}a)\,||\boldsymbol{z}||\,x_t}{\sqrt{||\cosh(\sqrt{-K}a)\boldsymbol{z}||^2 - (\sinh(\sqrt{-K}a)||\boldsymbol{z}||)^2}} \right) \right|. \tag{10}$$

**MLR in the Lorentz model** Lebanon & Lafferty (2004) formulated the logits of the Euclidean MLR classifier using the distance from instances to hyperplanes describing the class regions. Specifically, given input $\boldsymbol{x} \in \mathbb{R}^n$ and $C$ classes, the output probability of class $c \in \{1, ..., C\}$ can be expressed as

$$p(y = c \mid \boldsymbol{x}) \propto \exp(v_{\boldsymbol{w}_c}(\boldsymbol{x})), \quad v_{\boldsymbol{w}_c}(\boldsymbol{x}) = \mathrm{sign}(\langle \boldsymbol{w}_c, \boldsymbol{x} \rangle)||\boldsymbol{w}_c||d(\boldsymbol{x}, H_{\boldsymbol{w}_c}), \quad \boldsymbol{w}_c \in \mathbb{R}^n, \tag{11}$$

where $H_{\boldsymbol{w}_c}$ is the decision hyperplane of class $c$.

We define the Lorentz MLR without loss of generality by inserting the Lorentzian counterparts into Eq. 11. This yields logits given by the following theorem, proven in Appendix B.3.

**Theorem 2** *Given parameters $a_c \in \mathbb{R}$ and $\boldsymbol{z}_c \in \mathbb{R}^n$, the Lorentz MLR's output logit corresponding to class $c$ and input $\boldsymbol{x} \in \mathbb{L}_K^n$ is given by*

$$v_{\boldsymbol{z}_c, a_c}(\boldsymbol{x}) = \frac{1}{\sqrt{-K}} \mathrm{sign}(\alpha)\beta \left| \sinh^{-1} \left( \sqrt{-K} \frac{\alpha}{\beta} \right) \right|, \tag{12}$$

$$\alpha = \cosh(\sqrt{-K}a)\langle \boldsymbol{z}, \boldsymbol{x}_s \rangle - \sinh(\sqrt{-K}a),$$

$$\beta = \sqrt{||\cosh(\sqrt{-K}a)\boldsymbol{z}||^2 - (\sinh(\sqrt{-K}a)||\boldsymbol{z}||)^2}.$$

## 4.4 LORENTZ RESIDUAL CONNECTION AND ACTIVATION

**Residual connection** The residual connection is a crucial component when designing deep CNNs. As vector addition is ill-defined in the Lorentz model, we add the vector's space components and concatenate a corresponding time component. This is possible as a point $\boldsymbol{x} \in \mathbb{L}_K^n$ can be defined by an arbitrary space component $\boldsymbol{x}_s \in \mathbb{R}^n$ and a time component $x_t = \sqrt{||\boldsymbol{x}_s||^2 - 1/K}$. Our method is straightforward and provides the best empirical performance compared to other viable methods for addition we implemented, i.e., tangent space addition (Nickel & Kiela, 2018), parallel transport addition (Chami et al., 2019), Möbius addition (after projecting to the Poincaré ball) (Ganea et al., 2018), and fully-connected layer addition (Chen et al., 2021).

**Non-linear activation** Prior works use non-linear activation in tangent space (Fan et al., 2022), which weakens the model's stability due to frequent logarithmic and exponential maps. We propose a simpler operation for the Lorentz model by applying the activation function to the space component and concatenating a time component. For example, the Lorentz ReLU activation is given by

$$\boldsymbol{y} = \begin{bmatrix} \sqrt{||\mathrm{ReLU}(\boldsymbol{x}_s)||^2 - 1/K} \\ \mathrm{ReLU}(\boldsymbol{x}_s) \end{bmatrix}. \tag{13}$$

## 5 EXPERIMENTS

We evaluate hyperbolic models on image classification and generation tasks and compare them against Euclidean and hybrid HNN counterparts from the literature. To ensure a fair comparison, in every task, we directly translate a Euclidean baseline to the hyperbolic setting by using hyperbolic modules as one-to-one replacements. All experiments are implemented in PyTorch (Paszke et al., 2019), and we optimize hyperbolic models using adaptive Riemannian optimizers (Bécigneul & Ganea, 2018) provided by Geoopt (Kochurov et al., 2020), with floating-point precision set to 32 bits. We provide detailed experimental configurations in Appendix C and ablation experiments in Appendix D.

### 5.1 IMAGE CLASSIFICATION

**Experimental setup**  We evaluate image classification performance using ResNet-18 (He et al., 2015b) and three datasets: CIFAR-10 (Krizhevsky, 2009), CIFAR-100 (Krizhevsky, 2009), and Tiny-ImageNet (Le & Yang, 2015). All these datasets exhibit hierarchical class relations and high hyperbolicity (low $\delta_{rel}$), making the use of hyperbolic models well-motivated.

For the HCNN, we replace all components in the ResNet architecture with our proposed Lorentz modules. Additionally, we experiment with a novel hybrid approach (HECNN), where we employ our Lorentz decoder and replace only the ResNet encoder blocks with the highest hyperbolicity ($\delta_{rel} < 0.2$), i.e., blocks 1 and 3 (see Appendix D.1). To establish hyperbolic baselines we follow the literature (Atigh et al., 2022; Guo et al., 2022) and implement hybrid HNNs with a Euclidean encoder and a hyperbolic output layer (using both the Poincaré MLR (Shimizu et al., 2020) and our novel Lorentz MLR). Additionally, we report classification results for the concurrently developed fully hyperbolic Poincaré ResNet (van Spengler et al., 2023). For all models, we adopt the training procedure and hyperparameters of DeVries & Taylor (2017), which have been optimized for Euclidean CNNs and yield a strong Euclidean ResNet baseline.

**Main results**  Table 1 shows that hyperbolic models using the Lorentz model achieve the highest accuracy across all datasets, outperforming both the Euclidean and Poincaré baselines. In contrast, the Poincaré HNNs are consistently worse than the Euclidean baseline, aligning with the results of Guo et al. (2022). Notably, only in the case of CIFAR-10, all models exhibit equal performance, which is expected due to the dataset's simplicity. We also notice that the hybrid encoder model outperforms the fully hyperbolic model, indicating that not all parts of the model benefit from hyperbolic geometry. Overall, our findings suggest that the Lorentz model is better suited for HNNs than the Poincaré ball. This may be attributed to the better numerical stability causing fewer inaccuracies (Mishne et al., 2022). Furthermore, we achieve a notable improvement (of up to 1.5%) in the accuracy of current HNNs. This shows the potential of using our HCNN components in advancing HNNs.

Table 1: Classification accuracy (%) of ResNet-18 models. We estimate the mean and standard deviation from five runs. The best performance is highlighted in bold (higher is better).

| | CIFAR-10 ($\delta_{rel} = 0.26$) | CIFAR-100 ($\delta_{rel} = 0.23$) | Tiny-ImageNet ($\delta_{rel} = 0.20$) |
|---|---|---|---|
| Euclidean (He et al., 2015b) | $95.14_{\pm 0.12}$ | $77.72_{\pm 0.15}$ | $65.19_{\pm 0.12}$ |
| Hybrid Poincaré (Guo et al., 2022) | $95.04_{\pm 0.13}$ | $77.19_{\pm 0.50}$ | $64.93_{\pm 0.38}$ |
| Hybrid Lorentz (Ours) | $94.98_{\pm 0.12}$ | $78.03_{\pm 0.21}$ | $65.63_{\pm 0.10}$ |
| Poincaré ResNet (van Spengler et al., 2023) | $94.51_{\pm 0.15}$ | $76.60_{\pm 0.32}$ | $62.01_{\pm 0.56}$ |
| HECNN Lorentz (Ours) | $\mathbf{95.16_{\pm 0.11}}$ | $\mathbf{78.76_{\pm 0.24}}$ | $\mathbf{65.96_{\pm 0.18}}$ |
| HCNN Lorentz (Ours) | $95.14_{\pm 0.08}$ | $78.07_{\pm 0.17}$ | $65.71_{\pm 0.13}$ |

**Adversarial robustness**  Prior works have demonstrated the robustness of hyperbolic models against adversarial attacks (Yue et al., 2023; Guo et al., 2022). We expect better performance for HCNNs/HECNNs due to the bigger effect fully hyperbolic models have on the embedding space as can be seen in Figure 3. We believe the benefit could come from the increased inter-class separation afforded by the distance metric which allows for greater slack in the object classification. To study this, we employ the trained models and attack them using FGSM (Goodfellow et al., 2015) and PGD (Madry et al., 2019) with different perturbations. The results in Table 2 show that our HCNN is more

Table 2: Classification accuracy (%) after performing FGSM and PGD attacks on CIFAR-100. We estimate the mean and standard deviation from attacking five trained models (higher is better).

| | FGSM | | | PGD | | |
|---|---|---|---|---|---|---|
| Max. perturbation $\epsilon$ | 0.8/255 | 1.6/255 | 3.2/255 | 0.8/255 | 1.6/255 | 3.2/255 |
| Euclidean (He et al., 2015b) | $65.70_{\pm 0.28}$ | $54.98_{\pm 0.39}$ | $39.97_{\pm 0.43}$ | $64.43_{\pm 0.29}$ | $49.76_{\pm 0.42}$ | $26.30_{\pm 0.40}$ |
| Hybrid Poincaré (Guo et al., 2022) | $64.68_{\pm 0.40}$ | $53.32_{\pm 0.60}$ | $37.52_{\pm 0.50}$ | $63.43_{\pm 0.44}$ | $48.41_{\pm 0.60}$ | $23.78_{\pm 0.75}$ |
| Hybrid Lorentz (Ours) | $65.27_{\pm 0.52}$ | $53.82_{\pm 0.49}$ | $40.53_{\pm 0.31}$ | $64.15_{\pm 0.53}$ | $49.05_{\pm 0.68}$ | $27.17_{\pm 0.40}$ |
| HECNN Lorentz (Ours) | $66.13_{\pm 0.41}$ | $55.71_{\pm 0.43}$ | $42.76_{\pm 0.37}$ | $65.01_{\pm 0.49}$ | $50.82_{\pm 0.37}$ | $30.34_{\pm 0.22}$ |
| HCNN Lorentz (Ours) | $\mathbf{66.47_{\pm 0.27}}$ | $\mathbf{57.14_{\pm 0.30}}$ | $\mathbf{43.51_{\pm 0.35}}$ | $\mathbf{65.04_{\pm 0.28}}$ | $\mathbf{52.25_{\pm 0.34}}$ | $\mathbf{31.77_{\pm 0.55}}$ |

robust, achieving up to 5% higher accuracy. In addition, and contrary to Guo et al. (2022), we observe that hybrid decoder HNNs can be more susceptible to adversarial attacks than Euclidean models.

**Low embedding dimensionality** HNNs have shown to be most effective for low-dimensional embeddings (Peng et al., 2022). To this end, we reduce the dimensionality of the final ResNet block and the embeddings and evaluate classification accuracy on CIFAR-100.

The results in Figure 3 verify the effectiveness of hyperbolic spaces with low dimensions, where all HNNs outperform the Euclidean models. However, our HCNN and HECNN can leverage this advantage best, suggesting that hyperbolic encoders offer great opportunities for dimensionality reduction and designing smaller mod-

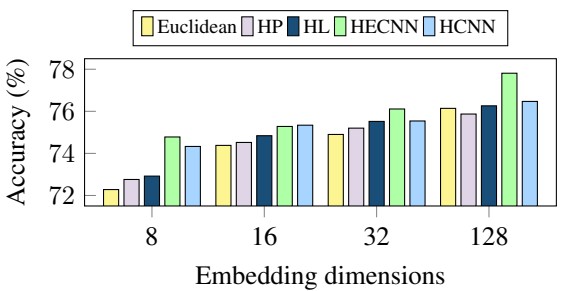

Figure 3: CIFAR-100 accuracy obtained with lower dimensionalities in the final ResNet block.

els with fewer parameters. The high performance of HECNN is unexpected as we hypothesized the fully hyperbolic model to perform best. This implies that hybrid encoder HNNs might make better use of the combined characteristics of both Euclidean and hyperbolic spaces.

## 5.2 IMAGE GENERATION

**Experimental setup** Variational autoencoders (VAEs) (Kingma & Welling, 2013; Rezende et al., 2014) have been widely adopted in HNN research to model latent embeddings in hyperbolic spaces (Nagano et al., 2019; Mathieu et al., 2019; Ovinnikov, 2019; Hsu et al., 2020). HNNs have shown to generate more expressive embeddings under lower dimensionalities which would make them a good fit for VAEs. In this experiment, we extend the hyperbolic VAE to the fully hyperbolic setting using our proposed HCNN framework and, for the first time, evaluate its performance on image generation using the standard Fréchet Inception Distance (FID) metric (Heusel et al., 2017).

Building on the experimental setting of Ghosh et al. (2019), we test vanilla VAEs and assess generative performance on CIFAR-10 (Krizhevsky, 2009), CIFAR-100 (Krizhevsky, 2009), and CelebA (Liu et al., 2015) datasets. We compare our HCNN-VAE against the Euclidean and two hybrid models. Following prior works, the hybrid models only include a latent hyperbolic distribution and no hyperbolic layers. Specifically, we employ the wrapped normal distributions in the Lorentz model (Nagano et al., 2019) and the Poincaré ball (Mathieu et al., 2019), respectively.

**Main results** The results in Table 3 show that our HCNN-VAE outperforms all baselines. Likewise, the hybrid models improve performance over the Euclidean model, indicating that learning the latent embeddings in hyperbolic spaces is beneficial. This is likely due to the higher representation capacity of the hyperbolic space, which is crucial in low dimensional settings. However, our HCNN is better at leveraging the advantages of hyperbolic geometry due to its fully hyperbolic architecture. These results suggest that our method is a promising approach for generation and for modeling latent structures in image data.

Table 3: Reconstruction and generation FID of manifold VAEs across five runs (lower is better).

| | CIFAR-10 | | CIFAR-100 | | CelebA | |
|---|---|---|---|---|---|---|
| | Rec. FID | Gen. FID | Rec. FID | Gen. FID | Rec. FID | Gen. FID |
| Euclidean | $61.21_{\pm 0.72}$ | $92.40_{\pm 0.80}$ | $63.81_{\pm 0.47}$ | $103.54_{\pm 0.84}$ | $54.80_{\pm 0.29}$ | $79.25_{\pm 0.89}$ |
| Hybrid Poincaré (Mathieu et al., 2019) | $59.85_{\pm 0.50}$ | $90.13_{\pm 0.77}$ | $62.64_{\pm 0.43}$ | $\mathbf{98.19_{\pm 0.57}}$ | $54.62_{\pm 0.61}$ | $81.30_{\pm 0.56}$ |
| Hybrid Lorentz (Nagano et al., 2019) | $59.29_{\pm 0.47}$ | $90.91_{\pm 0.84}$ | $62.14_{\pm 0.35}$ | $98.34_{\pm 0.62}$ | $54.64_{\pm 0.34}$ | $82.78_{\pm 0.93}$ |
| HCNN Lorentz (Ours) | $\mathbf{57.78_{\pm 0.56}}$ | $\mathbf{89.20_{\pm 0.85}}$ | $\mathbf{61.44_{\pm 0.64}}$ | $100.27_{\pm 0.84}$ | $\mathbf{54.17_{\pm 0.66}}$ | $\mathbf{78.11_{\pm 0.95}}$ |

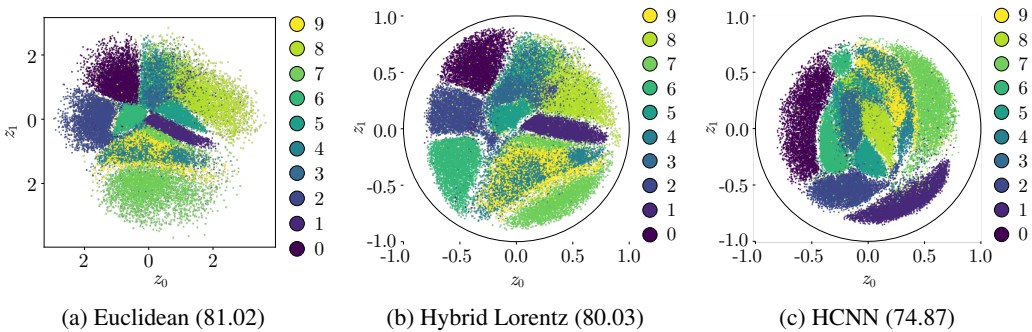

(a) Euclidean (81.02)     (b) Hybrid Lorentz (80.03)     (c) HCNN (74.87)

Figure 4: Embeddings of MNIST dataset in 2D latent space of VAEs (with gen. FID). Colors represent golden labels and Lorentz embeddings are projected onto the Poincaré ball for better visualization.

**Analysis of latent embeddings** The latent embedding space is a crucial component of VAEs as it influences how the data's features are encoded and used for generating the output. We visually analyze the distribution of latent embeddings inferred by the VAEs. For this, the models are retrained on the MNIST (Lecun et al., 1998) dataset with an embedding dimension $d_E = 2$. Then, the images of the training dataset are passed through the encoder and visualized as shown in Figure 4.

We observe the formation of differently shaped clusters that correlate with the ground truth labels. While the embeddings of the Euclidean and hybrid models form many clusters that direct towards the origin, the HCNN-VAE obtains rather curved clusters that maintain a similar distance from the origin. The structures within the HCNN's latent space can be interpreted as hierarchies where the distance to the origin represents hierarchical levels. As these structures cannot be found for the hybrid model, our results suggest that hybrid HNNs using only a single hyperbolic layer have little impact on the model's Euclidean characteristics. Conversely, our fully hyperbolic architecture significantly impacts how features are represented and learned, directing the model toward tree-like structures.

## 6 CONCLUSION

In this work, we proposed HCNN, a generalization of the convolutional neural network that learns latent feature representations in hyperbolic spaces. To this end, we formalized the necessary modules in the Lorentz model, deriving novel formulations of the convolutional layer, batch normalization, and multinomial logistic regression. We empirically demonstrated that ResNet and VAE models based on our hyperbolic framework achieve better performance on standard vision tasks than Euclidean and hybrid decoder baselines, especially in adversarial and lower dimensional settings. Additionally, we showed that using the Lorentz model in HNNs leads to better stability and performance than the Poincaré ball.

However, hyperbolic CNNs are still in their early stages and introduce mathematical complexity and computational overhead. For this, we explored HECNN models with the benefit of targeting only specific parts of the encoder, allowing for faster runtimes and larger models. Moreover, our framework currently relies on generalizations of neural network layers that were designed for Euclidean geometry and might not fully capture the unique properties of hyperbolic geometry. Further research is needed to fully understand the properties of HCNNs and address open questions such as optimization, scalability, and performance on other deep learning problems. We hope our work will inspire future research and development in this exciting and rapidly evolving field.

ACKNOWLEDGMENTS

This work was performed on the HoreKa supercomputer funded by the Ministry of Science, Research and the Arts Baden-Württemberg and by the Federal Ministry of Education and Research. Ahmad Bdeir and Kristian Schwethelm were funded by the European Union's Horizon 2020 research and innovation programme under the SustInAfrica grant agreement No 861924. Kristian Schwethelm was also funded by the Deutsche Forschungsgemeinschaft (DFG, German Research Foundation) - project number 225197905.

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

# A OPERATIONS IN HYPERBOLIC GEOMETRY

## A.1 LORENTZ AND POINCARE

**Poincaré ball** The n-dimensional Poincaré ball $\mathbb{B}_K^n = (\mathcal{B}^n, \mathfrak{g}_{\boldsymbol{x}}^K)$ is defined by $\mathcal{B}^n = \{\boldsymbol{x} \in \mathbb{R}^n \mid -K\|\boldsymbol{x}\|^2 < 1\}$ and the Riemannian metric $\mathfrak{g}_{\boldsymbol{x}}^K = (\lambda_{\boldsymbol{x}}^K)^2 \mathbf{I}_n$, where $\lambda_{\boldsymbol{x}}^K = 2(1 + K\|\boldsymbol{x}\|^2)^{-1}$. It describes the hyperbolic space by an open ball of radius $\sqrt{-1/K}$, see Figure 2.

## A.2 LORENTZ MODEL

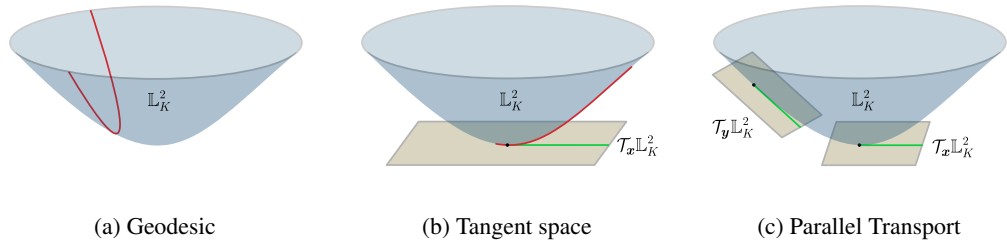

(a) Geodesic      (b) Tangent space      (c) Parallel Transport

Figure 5: Illustrations of geometrical operations in the 2-dimensional Lorentz model. (a) The shortest distance between two points is represented by the connecting geodesic (red line). (b) The red line gets projected onto the tangent space of the origin resulting in the green line. (c) The green line gets parallel transported to the tangent space of the origin.

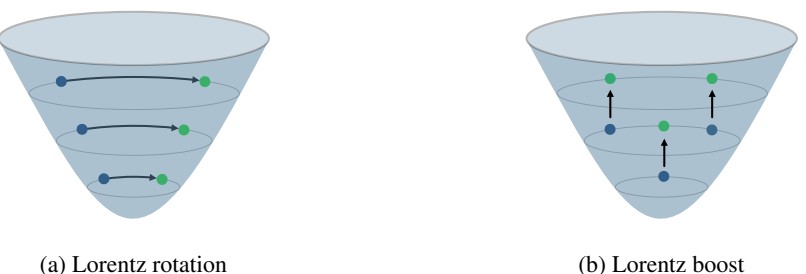

(a) Lorentz rotation      (b) Lorentz boost

Figure 6: Illustration of the Lorentz transformations in the 2-dimensional Lorentz model.

In this section, we describe essential geometrical operations in the Lorentz model. Most of these operations are defined for all Riemannian manifolds and thus introduced for the general case first. However, the closed-form formulae are only given for the Lorentz model. We also provide visual illustrations in Figure 5.

**Distance** Distance is defined as the length of the shortest path between a pair of points on a surface. While in Euclidean geometry, this is a straight line, in hyperbolic space, the shortest path is represented by a curved geodesic generalizing the notion of a straight line. In the Lorentz model, the distance is inherited from Minkowski space. Let $\boldsymbol{x}, \boldsymbol{y} \in \mathbb{L}_K^n$ denote two points in the Lorentz model. Then, the length of the connecting geodesic and, thereby, the distance is given by

$$d_{\mathcal{L}}(\boldsymbol{x}, \boldsymbol{y}) = \frac{1}{\sqrt{-K}} \cosh^{-1}(K \langle \boldsymbol{x}, \boldsymbol{y} \rangle_{\mathcal{L}}), \tag{14}$$

and the squared distance (Law et al., 2019) by

$$d_{\mathcal{L}}^2(\boldsymbol{x}, \boldsymbol{y}) = \|\boldsymbol{x} - \boldsymbol{y}\|_{\mathcal{L}}^2 = \frac{2}{K} - 2 \langle \boldsymbol{x}, \boldsymbol{y} \rangle_{\mathcal{L}}. \tag{15}$$

When calculating the distance of any point $\boldsymbol{x} \in \mathbb{L}_K^n$ to the origin $\overline{\mathbf{0}}$, the equations can be simplified to

$$d_{\mathcal{L}}(\boldsymbol{x}, \overline{\mathbf{0}}) = ||\log_{\overline{\mathbf{0}}}^K(\boldsymbol{x})||, \tag{16}$$

$$d_{\mathcal{L}}^2(\boldsymbol{x}, \overline{\mathbf{0}}) = \frac{2}{K}(1 + \sqrt{-K}x_t). \tag{17}$$

**Tangent space** The space around each point $\boldsymbol{x}$ on a differentiable manifold $\mathcal{M}$ can be linearly approximated by the tangent space $\mathcal{T}_{\boldsymbol{x}}\mathcal{M}$. It is a first-order approximation bridging the gap to Euclidean space. This helps performing Euclidean operations, but it introduces an approximation error, which generally increases with the distance from the reference point. Let $\boldsymbol{x} \in \mathbb{L}_K^n$, then the tangent space at point $\boldsymbol{x}$ can be expressed as

$$\mathcal{T}_{\boldsymbol{x}}\mathbb{L}_K^n := \{\boldsymbol{y} \in \mathbb{R}^{n+1} \mid \langle \boldsymbol{y}, \boldsymbol{x} \rangle_{\mathcal{L}} = 0\}. \tag{18}$$

**Exponential and logarithmic maps** Exponential and logarithmic maps are mappings between the manifold $\mathcal{M}$ and the tangent space $\mathcal{T}_{\boldsymbol{x}}\mathcal{M}$ with $\boldsymbol{x} \in \mathcal{M}$. The exponential map $\exp_{\boldsymbol{x}}^K(\boldsymbol{z}) : \mathcal{T}_{\boldsymbol{x}}\mathbb{L}_K^n \to \mathbb{L}_K^n$ maps a tangent vector $\boldsymbol{z} \in \mathcal{T}_{\boldsymbol{x}}\mathbb{L}_K^n$ on the Lorentz manifold by

$$\exp_{\boldsymbol{x}}^K(\boldsymbol{z}) = \cosh(\alpha)\boldsymbol{x} + \sinh(\alpha)\frac{\boldsymbol{z}}{\alpha}, \text{ with } \alpha = \sqrt{-K}||\boldsymbol{z}||_{\mathcal{L}}, \ ||\boldsymbol{z}||_{\mathcal{L}} = \sqrt{\langle \boldsymbol{z}, \boldsymbol{z} \rangle_{\mathcal{L}}}. \tag{19}$$

The logarithmic map is the inverse mapping and maps a vector $\boldsymbol{y} \in \mathbb{L}_K^n$ to the tangent space of $\boldsymbol{x}$ by

$$\log_{\boldsymbol{x}}^K(\boldsymbol{y}) = \frac{\cosh^{-1}(\beta)}{\sqrt{\beta^2 - 1}} \cdot (\boldsymbol{y} - \beta\boldsymbol{x}), \text{ with } \beta = K\langle \boldsymbol{x}, \boldsymbol{y} \rangle_{\mathcal{L}}. \tag{20}$$

In the special case of working with the tangent space at the origin $\overline{\mathbf{0}}$, the exponential map simplifies to

$$\exp_{\overline{\mathbf{0}}}^K(\boldsymbol{z}) = \frac{1}{\sqrt{-K}}\left[\cosh(\sqrt{-K}||\boldsymbol{z}||), \ \sinh(\sqrt{-K}||\boldsymbol{z}||)\frac{\boldsymbol{z}}{||\boldsymbol{z}||}\right]. \tag{21}$$

**Parallel transport** The parallel transport operation $\mathrm{PT}_{\boldsymbol{x} \to \boldsymbol{y}}^K(\boldsymbol{v})$ maps a vector $\boldsymbol{v} \in \mathcal{T}_{\boldsymbol{x}}\mathcal{M}$ from the tangent space of $\boldsymbol{x} \in \mathcal{M}$ to the tangent space of $\boldsymbol{y} \in \mathcal{M}$. It preserves the local geometry around the reference point by moving the points along the geodesic connecting $\boldsymbol{x}$ and $\boldsymbol{y}$. The formula for the Lorentz model is given by

$$\mathrm{PT}_{\boldsymbol{x} \to \boldsymbol{y}}^K(\boldsymbol{v}) = \boldsymbol{v} - \frac{\langle \log_{\boldsymbol{x}}^K(\boldsymbol{y}), \boldsymbol{v} \rangle_{\mathcal{L}}}{d_{\mathcal{L}}(\boldsymbol{x}, \boldsymbol{y})}(\log_{\boldsymbol{x}}^K(\boldsymbol{y}) + \log_{\boldsymbol{y}}^K(\boldsymbol{x})) \tag{22}$$

$$= \boldsymbol{v} + \frac{\langle \boldsymbol{y}, \boldsymbol{v} \rangle_{\mathcal{L}}}{\frac{1}{-K} - \langle \boldsymbol{x}, \boldsymbol{y} \rangle_{\mathcal{L}}}(\boldsymbol{x} + \boldsymbol{y}). \tag{23}$$

**Lorentzian centroid (Law et al., 2019)** The weighted centroid with respect to the squared Lorentzian distance, which solves $\min_{\boldsymbol{\mu} \in \mathbb{L}_K^n} \sum_{i=1}^m \nu_i d_{\mathcal{L}}^2(\boldsymbol{x}_i, \boldsymbol{\mu})$, with $\boldsymbol{x}_i \in \mathbb{L}_K^n$ and $\nu_i \geq 0, \sum_{i=1}^m \nu_i > 0$, is given by

$$\boldsymbol{\mu} = \frac{\sum_{i=1}^m \nu_i \boldsymbol{x}_i}{\sqrt{-K}\left|||\sum_{i=1}^m \nu_i \boldsymbol{x}_i||_{\mathcal{L}}\right|}. \tag{24}$$

**Lorentz average pooling** The average pooling layer is implemented by computing the Lorentzian centroid of all hyperbolic features within the receptive field.

**Lorentz transformations** The set of linear transformations in the Lorentz model are called Lorentz transformations. A transformation matrix $\mathbf{A}^{(n+1)\times(n+1)}$ that linearly maps $\mathbb{R}^{n+1} \to \mathbb{R}^{n+1}$ is called Lorentz transformation if and only if $\langle \mathbf{A}\boldsymbol{x}, \mathbf{A}\boldsymbol{y} \rangle_{\mathcal{L}} = \langle \boldsymbol{x}, \boldsymbol{y} \rangle_{\mathcal{L}} \ \forall \ \boldsymbol{x}, \boldsymbol{y} \in \mathbb{R}^{n+1}$. The set of matrices forms an orthogonal group $\boldsymbol{O}(1, n)$ called the Lorentz group. As the Lorentz model only uses the upper sheet of the two-sheeted hyperboloid, the transformations under consideration here lie within the positive Lorentz group $\boldsymbol{O}^+(1, n) = \{\mathbf{A} \in \boldsymbol{O}(1, n) : a_{11} > 0\}$, preserving the sign of the time component $x_t$ of $\boldsymbol{x} \in \mathbb{L}_K^n$. Specifically, here, the Lorentz transformations can be formulated as

$$\boldsymbol{O}^+(1, n) = \{\mathbf{A} \in \mathbb{R}^{(n+1)\times(n+1)} \mid \forall \boldsymbol{x} \in \mathbb{L}_K^n : \langle \mathbf{A}\boldsymbol{x}, \mathbf{A}\boldsymbol{x} \rangle_{\mathcal{L}} = \frac{1}{K}, (\mathbf{A}\boldsymbol{x})_0 > 0)\}. \tag{25}$$

Any Lorentz transformation can be decomposed into a Lorentz rotation and Lorentz boost by polar decomposition $\mathbf{A} = \mathbf{RB}$ (Moretti, 2002). The former rotates points around the time axis, using matrices given by

$$\mathbf{R} = \left[ \begin{array}{cc} 1 & \mathbf{0}^T \\ \mathbf{0} & \tilde{\mathbf{R}} \end{array} \right], \tag{26}$$

where $\mathbf{0}$ is a zero vector, $\tilde{\mathbf{R}}^T \tilde{\mathbf{R}} = \mathbf{I}$, and $\det(\tilde{\mathbf{R}}) = 1$. This shows that the Lorentz rotations for the upper sheet lie in a special orthogonal subgroup $\boldsymbol{SO}^+(1, n)$ preserving the orientation, while $\tilde{\mathbf{R}} \in \boldsymbol{SO}(n)$. On the other side, the Lorentz boost moves points along the spatial axis given a velocity $\boldsymbol{v} \in \mathbb{R}^n, ||\boldsymbol{v}|| < 1$ without rotating them along the time axis. Formally, the boost matrices are given by

$$\mathbf{B} = \left[ \begin{array}{cc} \gamma & -\gamma\boldsymbol{v}^T \\ -\gamma\boldsymbol{v} & \mathbf{I} + \frac{\gamma^2}{1+\gamma}\boldsymbol{v}\boldsymbol{v}^T \end{array} \right], \tag{27}$$

with $\gamma = \frac{1}{\sqrt{1-||\boldsymbol{v}||^2}}$. See Figure 6 for illustrations of the Lorentz rotation and Lorentz boost.

**Lorentz fully-connected layer** Recently, Chen et al. (2021) showed that the linear transformations performed in the tangent space (Ganea et al., 2018; Nickel & Kiela, 2018) can not apply all Lorentz transformations but only a special rotation and no boost. They proposed a direct method in pseudo-hyperbolic space[1], which can apply all Lorentz transformations. Specifically, let $\boldsymbol{x} \in \mathbb{L}_K^n$ denote the input vector and $\mathbf{W} \in \mathbb{R}^{m \times n+1}, \boldsymbol{v} \in \mathbb{R}^{n+1}$ the weight parameters, then the transformation matrix is given by

$$f_{\boldsymbol{x}}(\mathbf{M}) = f_{\boldsymbol{x}}\left( \left[ \begin{array}{c} \boldsymbol{v}^T \\ \mathbf{W} \end{array} \right] \right) = \left[ \begin{array}{c} \frac{\sqrt{||\mathbf{W}\boldsymbol{x}||^2 - 1/K}}{\boldsymbol{v}^T\boldsymbol{x}}\boldsymbol{v}^T \\ \mathbf{W} \end{array} \right]. \tag{28}$$

Adding other components of fully-connected layers, including normalization, the final definition of the proposed Lorentz fully-connected layer becomes

$$\boldsymbol{y} = \left[ \begin{array}{c} \sqrt{||\phi(\mathbf{W}\boldsymbol{x}, \boldsymbol{v})||^2 - 1/K} \\ \phi(\mathbf{W}\boldsymbol{x}, \boldsymbol{v}) \end{array} \right], \tag{29}$$

with operation function

$$\phi(\mathbf{W}\boldsymbol{x}, \boldsymbol{v}) = \lambda\sigma(\boldsymbol{v}^T\boldsymbol{x} + b')\frac{\mathbf{W}\psi(\boldsymbol{x}) + \boldsymbol{b}}{||\mathbf{W}\psi(\boldsymbol{x}) + \boldsymbol{b}||}, \tag{30}$$

where $\lambda > 0$ is a learnable scaling parameter and $\boldsymbol{b} \in \mathbb{R}^n, \psi, \sigma$ denote the bias, activation, and sigmoid function, respectively.

---

[1] Chen et al. (2021) note that their general formula is not fully hyperbolic, but a relaxation in implementation, while the input and output are still guaranteed to lie in the Lorentz model.

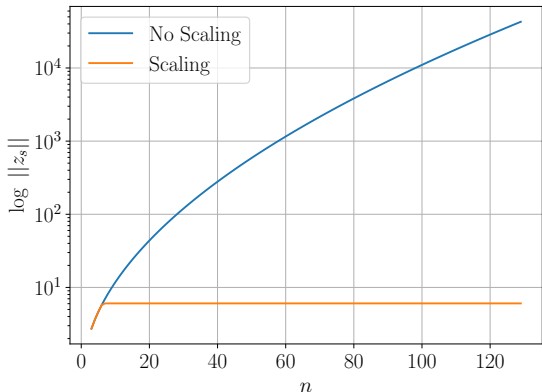

Figure 7: Growth of the space component's norm $||\boldsymbol{z}_s||$ after applying the exponential map to an $n$-dimensional vector $\mathbf{1}_n = (1,...,1) \in \mathcal{T}_{\overline{\mathbf{0}}}\mathbb{L}_K^n$ with curvature $K = -1$. The y-axis is scaled logarithmically.

In this work, we simplify the layer definition by removing the internal normalization, as we use batch normalization. This gives following formula for the Lorentz fully connected layer

$$
\boldsymbol{y} = \text{LFC}(\boldsymbol{x}) = \left[ \begin{array}{c} \sqrt{||\psi(\mathbf{W}\boldsymbol{x} + \boldsymbol{b})||^2 - 1/K} \\ \psi(\mathbf{W}\boldsymbol{x} + \boldsymbol{b}) \end{array} \right]. \tag{31}
$$

**Lorentz direct concatenation (Qu & Zou, 2022)**  Given a set of hyperbolic points $\{\boldsymbol{x}_i \in \mathbb{L}_K^n\}_{i=1}^N$, the Lorentz direct concatenation is given by

$$
\boldsymbol{y} = \text{HCat}(\{\boldsymbol{x}_i\}_{i=1}^N) = \left[ \sqrt{\sum_{i=1}^N x_{i_t}^2 + \frac{N-1}{K}}, \boldsymbol{x}_{1_s}^T, \ldots, \boldsymbol{x}_{N_s}^T \right]^T, \tag{32}
$$

where $\boldsymbol{y} \in \mathbb{L}_K^{nN} \subset \mathbb{R}^{nN+1}$.

**Wrapped normal distribution**  Nagano et al. (2019) proposed a wrapped normal distribution in the Lorentz model, which offers efficient sampling, great flexibility, and a closed-form density formulation. It can be constructed as follows:

1. Sample a Euclidean vector $\tilde{\boldsymbol{v}}$ from the Normal distribution $\mathcal{N}(\mathbf{0}, \boldsymbol{\Sigma})$.

2. Assume the sampled vector lies in the tangent space of the Lorentz model's origin $\boldsymbol{v} = [0, \tilde{\boldsymbol{v}}] \in \mathcal{T}_{\overline{\mathbf{0}}}\mathbb{L}_K^n$.

3. Parallel transport $\boldsymbol{v}$ from the tangent space of the origin to the tangent space of a new mean $\boldsymbol{\mu} \in \mathbb{L}_K^n$, yielding a tangent vector $\boldsymbol{u} \in \mathcal{T}_{\boldsymbol{\mu}}\mathbb{L}_K^n$.

4. Map $\boldsymbol{u}$ to $\mathbb{L}_K^n$ by applying the exponential map, yielding the final sample $\boldsymbol{z} \in \mathbb{L}_K^n$.

The distribution is parameterized by a Euclidean variance $\boldsymbol{\Sigma} \in \mathbb{R}^{n \times n}$ and a hyperbolic mean $\boldsymbol{\mu} \in \mathbb{L}_K^n$.

This method has shown to work well in hybrid HNN settings. However, in our fully hyperbolic VAE, high Euclidean variances destabilize the model. This is because, usually, the VAE's prior is set to a standard normal distribution with unit variance $\tilde{\boldsymbol{v}} \sim \mathcal{N}(\mathbf{0}, \mathbf{I})$. However, for high dimensional spaces, this leads to large values after the exponential map. That is why we propose to scale the prior variance as follows.

Let $\boldsymbol{v} \in \mathcal{T}_{\overline{\mathbf{0}}}\mathbb{L}_K^n$ denote a vector in the tangent space of the origin. Then the space component of the hyperbolic vector $\boldsymbol{z} \in \mathbb{L}_K^n$ resulting from the exponential map is given by

$$\boldsymbol{z}_s = (\exp_{\boldsymbol{0}}^K (\boldsymbol{v}))_s = \frac{1}{\sqrt{-K}} \sinh(\sqrt{-K}||\boldsymbol{v}||) \frac{\boldsymbol{v}}{||\boldsymbol{v}||}. \tag{33}$$

This shows that the norm of the space component depends on the sinh function, which grows approximately exponentially with the norm of the tangent vector ($||\boldsymbol{z}_s|| = \frac{1}{\sqrt{-K}} \sinh(\sqrt{-K}||\boldsymbol{v}||)$). The norm of the space component is important as it gets used to calculate the time component $z_t = \sqrt{||\boldsymbol{z}_s||^2 - 1/K}$, and it indicates how large the values of the hyperbolic points are. Now, assume an n-dimensional vector $\mathbf{1}_n = (1, ..., 1) \in \mathcal{T}_{\boldsymbol{0}}\mathbb{L}_K^n$, resembling the diagonal of the covariance matrix. Applying the exponential map to such a vector leads to fast-growing values with respect to the dimensionality $n$ because the norm of the tangent vector increases with $n$:

$$||\mathbf{1}_n|| = \sqrt{\sum_{i=1}^{n} 1^2} = \sqrt{n}. \tag{34}$$

To work against this, we propose to clip the norm of the prior variance as follows

$$\boldsymbol{\sigma}^2 = \begin{cases} \frac{s}{\sqrt{n}} : & \text{if } \sqrt{n} > s \\ \boldsymbol{\sigma}^2 : & \text{otherwise} \end{cases}, \tag{35}$$

where $s$ parameterizes the resulting norm. This achieves a clipped time component with respect to the dimensionality of $\mathbf{1}_n$ (see Figure 7). Furthermore, it offers nice interpretability using the Fréchet variance. As the distribution has zero mean, the Fréchet variance is given by the distance to the origin, which can be calculated by the norm of the tangent vector. This shows that this method controls the Fréchet variance. In practice, we empirically found $s = 2.5$ to be a good value.

Additionally, to prevent the HCNN-VAE from predicting relatively high variances, the scaling in Eq. 35 is applied. In this case, the Fréchet variance is not predefined, as usually $\boldsymbol{\sigma}^2 \neq \mathbf{1}_n$. However, it introduces a scaling operation resembling the variance scaling of the prior.

### A.3 Mapping between models

Because of the isometry between models of hyperbolic geometry, points in the Lorentz model can be mapped to the Poincaré ball by the following diffeomorphism

$$p_{\mathbb{L}_K^n \to \mathbb{B}_K^n}(\boldsymbol{x}) = \frac{\boldsymbol{x}_s}{x_t + \frac{1}{\sqrt{-K}}}. \tag{36}$$

## B Proofs

### B.1 Proofs for Lorentz hyperplane

This section contains the proof for the Euclidean reparameterization of the Lorentz hyperplane proposed by Mishne et al. (2022). Unfortunately, the authors only provided proof for the unit Lorentz model, i.e., assuming a curvature of $K = -1$. However, in general, the curvature can be different as $K < 0$. That is why we reproduce their proof for the general case.

**Proof for Eq. 8** Let $a \in \mathbb{R}$, $\boldsymbol{z} \in \mathbb{R}^n$, and $\overline{\boldsymbol{z}} \in \mathcal{T}_{\boldsymbol{0}}\mathbb{L}_K^n = [0, a\boldsymbol{z}/||\boldsymbol{z}||]$. Then, Mishne et al. (2022) parameterize a point in the Lorentz model as follows

$$\boldsymbol{p} \in \mathbb{L}_K^n := \exp_{\overline{\boldsymbol{0}}} \left( a \frac{\boldsymbol{z}}{||\boldsymbol{z}||} \right) \tag{37}$$

$$= \left[ \frac{1}{\sqrt{-K}} \cosh(\alpha), \ \sinh(\alpha) \frac{a \frac{\boldsymbol{z}}{||\boldsymbol{z}||}}{\alpha} \right]. \tag{38}$$

Now, with $\alpha = \sqrt{-K}||a\frac{z}{||z||}||_{\mathcal{L}} = \sqrt{-K}a$ we get

$$p = \left[\cosh(\sqrt{-K}a)\frac{1}{\sqrt{-K}}, \ \sinh(\sqrt{-K}a)\frac{a\frac{z}{||z||}}{\sqrt{-K}a}\right] \tag{39}$$

$$= \frac{1}{\sqrt{-K}}\left[\cosh(\sqrt{-K}a), \ \sinh(\sqrt{-K}a)\frac{z}{||z||}\right]. \tag{40}$$

This definition gets used to reparameterize the hyperplane parameter $w$ as follows

$$
\begin{aligned}
w &:= \mathrm{PT}^K_{\mathbf{0}\to p}(\overline{z})\\
&= \overline{z} + \frac{\langle p, \overline{z}\rangle_{\mathcal{L}}}{\frac{1}{-K} - \langle \mathbf{0}, p\rangle_{\mathcal{L}}}(\mathbf{0}+p)\\
&= [0, z]^T + \frac{\langle p, [0, z]^T\rangle_{\mathcal{L}}}{\frac{1}{-K} - \langle \mathbf{0}, p\rangle_{\mathcal{L}}}(\mathbf{0}+p)\\
&= [0, z]^T + \frac{\frac{1}{\sqrt{-K}}\sinh(\sqrt{-K}a)||z||}{\frac{1}{-K} + \frac{1}{-K}\cosh(\sqrt{-K}a)} \cdot \frac{1}{\sqrt{-K}}\left[1 + \cosh(\sqrt{-K}a), \ \sinh(\sqrt{-K}a)\frac{z}{||z||}\right]\\
&= [0, z]^T + \frac{\sinh(\sqrt{-K}a)||z||}{1 + \cosh(\sqrt{-K}a)} \cdot \left[1 + \cosh(\sqrt{-K}a), \ \sinh(\sqrt{-K}a)\frac{z}{||z||}\right]\\
&= \left[\sinh(\sqrt{-K}a)||z||, \ z + \frac{\sinh^2(\sqrt{-K}a)}{1 + \cosh(\sqrt{-K}a)}z\right]\\
&= \left[\sinh(\sqrt{-K}a)||z||, \ z + \frac{\cosh^2(\sqrt{-K}a) - 1}{1 + \cosh(\sqrt{-K}a)}z\right]\\
&= [\sinh(\sqrt{-K}a)||z||, \ \cosh(\sqrt{-K}a)z].
\end{aligned}
$$

**Proof for Eq. 9** After inserting Eq. 8 into Eq. 7 and solving the inner product, the hyperplane definition becomes

$$\tilde{H}_{z,a} = \{x \in \mathbb{L}^n_K \mid \cosh(\sqrt{-K}a)\langle z, x_s\rangle - \sinh(\sqrt{-K}a)\,||z||\,x_t = 0\}. \tag{41}$$

## B.2 Proof for distance to Lorentz hyperplane

**Proof for Theorem 1** To proof the distance of a point to hyperplanes in the Lorentz model, we follow the approach of Cho et al. (2019) and utilize the hyperbolic reflection. The idea is, that a hyperplane defines a reflection that interchanges two half-spaces. Therefore, the distance from a point $x \in \mathbb{L}^n_K$ to the hyperplane $H_{w,p}$ can be calculated by halving the distance to its reflection in the hyperplane $x \to y_w$

$$d_{\mathcal{L}}(x, H_{w,p}) = \frac{1}{2}d_{\mathcal{L}}(x, y_w). \tag{42}$$

The hyperbolic reflection is well-known in the literature (Grosek, 2008) and can be formulated as

$$y_w = x + \frac{2\langle w, x\rangle_{\mathcal{L}}w}{\langle w, w\rangle_{\mathcal{L}}}, \tag{43}$$

where $w$ is the perpendicular vector to the hyperplane and $\langle w, w\rangle_{\mathcal{L}} > 0$. Now, inserting Eq. 43 into Eq. 42 we can compute the distance to the hyperplane as follows

$$d_{\mathcal{L}}(\boldsymbol{x}, H_{\boldsymbol{w},\boldsymbol{p}}) = \frac{1}{2\sqrt{-K}} \cosh^{-1}(K \langle \boldsymbol{x}, \boldsymbol{y_w} \rangle_{\mathcal{L}})$$

$$= \frac{1}{2\sqrt{-K}} \cosh^{-1}(K \langle \boldsymbol{x}, \boldsymbol{x} + \frac{2 \langle \boldsymbol{w}, \boldsymbol{x} \rangle_{\mathcal{L}} \boldsymbol{w}}{\langle \boldsymbol{w}, \boldsymbol{w} \rangle_{\mathcal{L}}} \rangle_{\mathcal{L}})$$

$$= \frac{1}{2\sqrt{-K}} \cosh^{-1}(2K \langle \boldsymbol{x}, \boldsymbol{x} \rangle_{\mathcal{L}} + K \langle \boldsymbol{x}, \frac{\langle \boldsymbol{w}, \boldsymbol{x} \rangle_{\mathcal{L}} \boldsymbol{w}}{\langle \boldsymbol{w}, \boldsymbol{w} \rangle_{\mathcal{L}}} \rangle_{\mathcal{L}})$$

$$= \frac{1}{2\sqrt{-K}} \cosh^{-1}\left( K \frac{1}{K} + 2K \left( \frac{\langle \boldsymbol{w}, \boldsymbol{x} \rangle_{\mathcal{L}}}{\sqrt{\langle \boldsymbol{w}, \boldsymbol{w} \rangle_{\mathcal{L}}}} \right)^2 \right)$$

$$= \frac{1}{2\sqrt{-K}} \cosh^{-1}\left( 1 + 2 \left( \sqrt{-K} \frac{\langle \boldsymbol{w}, \boldsymbol{x} \rangle_{\mathcal{L}}}{\sqrt{\langle \boldsymbol{w}, \boldsymbol{w} \rangle_{\mathcal{L}}}} \right)^2 \right)$$

$$= \frac{1}{\sqrt{-K}} \left| \sinh^{-1}\left( \sqrt{-K} \frac{\langle \boldsymbol{w}, \boldsymbol{x} \rangle_{\mathcal{L}}}{||\boldsymbol{w}||_{\mathcal{L}}} \right) \right|,$$

which gives the final formula:

$$d_{\mathcal{L}}(\boldsymbol{x}, H_{\boldsymbol{w},\boldsymbol{p}}) = \frac{1}{\sqrt{-K}} \left| \sinh^{-1}\left( \sqrt{-K} \frac{\langle \boldsymbol{w}, \boldsymbol{x} \rangle_{\mathcal{L}}}{||\boldsymbol{w}||_{\mathcal{L}}} \right) \right|. \tag{44}$$

Comparing Eq. 44 to the equation of Cho et al. (2019) shows that the distance formula for hyperplanes in the unit Lorentz model can be extended easily to the general case by inserting the curvature parameter $K$ at two places.

Finally, defining $\boldsymbol{w}$ with the aforementioned reparameterization

$$\boldsymbol{w} := \mathrm{PT}_{\boldsymbol{0} \to \boldsymbol{p}}^{K}(\overline{\boldsymbol{z}}) = [\sinh(\sqrt{-K}a)||\boldsymbol{z}||, \cosh(\sqrt{-K}a)\boldsymbol{z}], \tag{45}$$

and solving the inner products, gives our final distance formula

$$d_{\mathcal{L}}(\boldsymbol{x}, \tilde{H}_{\boldsymbol{z},a}) = \frac{1}{\sqrt{-K}} \left| \sinh^{-1}\left( \sqrt{-K} \frac{\cosh(\sqrt{-K}a)\langle \boldsymbol{z}, \boldsymbol{x}_s \rangle - \sinh(\sqrt{-K}a) \, ||\boldsymbol{z}|| \, x_t}{\sqrt{||\cosh(\sqrt{-K}a)\boldsymbol{z}||^2 - (\sinh(\sqrt{-K}a)||\boldsymbol{z}||)^2}} \right) \right|. \tag{46}$$

### B.3 Proof for logits in the Lorentz MLR classifier

**Proof for Theorem 2** Following Lebanon & Lafferty (2004), given input $\boldsymbol{x} \in \mathbb{R}^n$ and $C$ classes, the Euclidean MLR logits of class $c \in \{1, ..., C\}$ can be expressed as

$$v_{\boldsymbol{w}_c}(\boldsymbol{x}) = \mathrm{sign}(\langle \boldsymbol{w}_c, \boldsymbol{x} \rangle)||\boldsymbol{w}_c|| d(x, H_{\boldsymbol{w}_c}), \quad \boldsymbol{w}_c \in \mathbb{R}^n, \tag{47}$$

where $H_{\boldsymbol{w}_c}$ is the decision hyperplane of class $c$.

Replacing the Euclidean operations with their counterparts in the Lorentz model yields logits of class $c$ for $\boldsymbol{x} \in \mathbb{L}_K^n$ as follows

$$v_{\boldsymbol{w}_c, \boldsymbol{p}_c}(\boldsymbol{x}) = \mathrm{sign}(\langle \boldsymbol{w}_c, \boldsymbol{x} \rangle_{\mathcal{L}})||\boldsymbol{w}_c||_{\mathcal{L}} d_{\mathcal{L}}(x, H_{\boldsymbol{w}_c, \boldsymbol{p}_c}), \tag{48}$$

with $\boldsymbol{w}_c \in \mathcal{T}_{\boldsymbol{p}_c} \mathbb{L}_K^n, \boldsymbol{p}_c \in \mathbb{L}_K^n$, and $\langle \boldsymbol{w}_c, \boldsymbol{w}_c \rangle_{\mathcal{L}} > 0$.

Inserting Eq. 44 into Eq. 48 gives a general formula without our reparameterization

$$v_{\boldsymbol{w}_c,\boldsymbol{p}_c}(\boldsymbol{x}) = \frac{1}{\sqrt{-K}} \operatorname{sign}(\langle \boldsymbol{w}_c, \boldsymbol{x} \rangle_{\mathcal{L}}) ||\boldsymbol{w}_c||_{\mathcal{L}} \left| \sinh^{-1}\left( \sqrt{-K} \frac{\langle \boldsymbol{w}_c, \boldsymbol{x} \rangle_{\mathcal{L}}}{||\boldsymbol{w}_c||_{\mathcal{L}}} \right) \right|. \tag{49}$$

Now, we reparameterize $\boldsymbol{w}$ with Eq. 45 again, which gives

$$\alpha := \langle \boldsymbol{w}_c, \boldsymbol{x} \rangle_{\mathcal{L}} = \cosh(\sqrt{-K}a)\langle \boldsymbol{z}, \boldsymbol{x}_s \rangle - \sinh(\sqrt{-K}a), \tag{50}$$

$$\beta := ||\boldsymbol{w}_c||_{\mathcal{L}} = \sqrt{||\cosh(\sqrt{-K}a)\boldsymbol{z}||^2 - (\sinh(\sqrt{-K}a)||\boldsymbol{z}||)^2}, \tag{51}$$

with $a \in \mathbb{R}$ and $\boldsymbol{z} \in \mathbb{R}^n$. Finally, we obtain the equation in Theorem 2:

$$v_{\boldsymbol{z}_c,a_c}(\boldsymbol{x}) = \frac{1}{\sqrt{-K}} \operatorname{sign}(\alpha)\beta \left| \sinh^{-1}\left( \sqrt{-K} \frac{\alpha}{\beta} \right) \right|. \tag{52}$$

## C  Additional experimental details

### C.1  Classification

**Datasets**  For classification, we employ the benchmark datasets CIFAR-10 (Krizhevsky, 2009), CIFAR-100 (Krizhevsky, 2009), and Tiny-ImageNet (Le & Yang, 2015). The CIFAR-10 and CIFAR-100 datasets each contain 60,000 $32 \times 32$ colored images from 10 and 100 different classes, respectively. We use the dataset split implemented in PyTorch, which includes 50,000 training images and 10,000 testing images. Tiny-ImageNet is a small subset of the ImageNet (Deng et al., 2009) dataset, with 100,000 images of 200 classes downsized to $64 \times 64$. Here, we use the official validation split for testing our models.

**Settings**  Table 4 summarizes the hyperparameters we adopt from DeVries & Taylor (2017) to train all classification models. Additionally, we use standard data augmentation methods in training, i.e., random mirroring and cropping. Regarding the feature clipping in hybrid HNNs, we tune the feature clipping parameter $r$ between 0.8 and 5.0 and find that, for most experiments, the best feature clipping parameter is $r = 1$. Only the Lorentz hybrid ResNet performs best with $r = 4$ on Tiny-ImageNet, and $r = 2$ on CIFAR-100 with lower embedding dimensions. Overall, we observe that the hybrid Lorentz ResNet has fewer gradient issues, allowing for higher clipping values. The HCNN-ResNet does not need tuning of any additional hyperparameters.

Table 4: Summary of hyperparameters used in training classification models.

| Hyperparameter | Value |
|---|---|
| Epochs | 200 |
| Batch size | 128 |
| Learning rate (LR) | 1e-1 |
| Drop LR epochs | 60, 120, 160 |
| Drop LR gamma | 0.2 |
| Weight decay | 5e-4 |
| Optimizer | (Riemannian)SGD |
| Floating point precision | 32 bit |
| GPU type | RTX A5000 |
| Num. GPUs | 1 or 2 |
| Hyperbolic curvature $K$ | $-1$ |

## C.2 GENERATION

**Datasets** For image generation, we use the aforementioned CIFAR-10 (Krizhevsky, 2009) and CIFAR-100 (Krizhevsky, 2009) datasets again. Additionally, we employ the CelebA (Liu et al., 2015) dataset, which includes colored $64 \times 64$ images of human faces. Here, we use the PyTorch implementation, containing 162,770 training images, 19,867 validation images, and 19,962 testing images.

**Settings** For hyperbolic and Euclidean models, we use the same architecture (see Table 5) and training hyperparameters (see Table 6). We employ a vanilla VAE similar to Ghosh et al. (2019) as the baseline Euclidean architecture (E-VAE). For the hybrid model, we replace the latent distribution of the E-VAE with the hyperbolic wrapped normal distribution in the Lorentz model (Nagano et al., 2019) and the Poincaré ball (Mathieu et al., 2019), respectively. Replacing all layers with our proposed hyperbolic counterparts yields the fully hyperbolic model. Here, we include the variance scaling mentioned in Section A.2, as otherwise training fails with NaN errors. Furthermore, we set the curvature $K$ for the Lorentz model to $-1$ and for the Poincaré ball to $-0.1$.

We evaluate the VAEs by employing two versions of the FID (Heusel et al., 2017) implemented by Seitzer (2020):

1. The *reconstruction FID* gives a lower bound on the generation quality. It is calculated by comparing test images with reconstructed validation images. As the CIFAR datasets have no official validation set, we exclude a fixed random portion of 10,000 images from the training set.

2. The *generation FID* measures the generation quality by comparing random generations from the models' latent space with the test set.

Table 5: Vanilla VAE architecture employed in all image generation experiments. Convolutional layers have a kernel size of $3 \times 3$ and transposed convolutional layers of $4 \times 4$. $s$ and $p$ denote stride and zero padding, respectively. The MLR in the Euclidean model is mimicked by a $1 \times 1$ convolutional layer.

| Layer | CIFAR-10/100 | CelebA |
|---|---|---|
| ENCODER: | | |
| $\rightarrow \text{PROJ}_{\mathbb{R}^n \rightarrow \mathbb{L}_K^n}$ | $32 \times 32 \times 3$ | $64 \times 64 \times 3$ |
| $\rightarrow \text{CONV}_{64, s2, p1} \rightarrow \text{BN} \rightarrow \text{RELU}$ | $16 \times 16 \times 64$ | $32 \times 32 \times 64$ |
| $\rightarrow \text{CONV}_{128, s2, p1} \rightarrow \text{BN} \rightarrow \text{RELU}$ | $8 \times 8 \times 128$ | $16 \times 16 \times 128$ |
| $\rightarrow \text{CONV}_{256, s2, p1} \rightarrow \text{BN} \rightarrow \text{RELU}$ | $4 \times 4 \times 256$ | $8 \times 8 \times 256$ |
| $\rightarrow \text{CONV}_{512, s2, p1} \rightarrow \text{BN} \rightarrow \text{RELU}$ | $2 \times 2 \times 512$ | $4 \times 4 \times 512$ |
| $\rightarrow \text{FLATTEN}$ | 2048 | 8192 |
| $\rightarrow \text{FC-MEAN}_d$ | 128 | 64 |
| $\rightarrow \text{FC-VAR}_d \rightarrow \text{SOFTPLUS}$ | 128 | 64 |
| DECODER: | | |
| $\rightarrow \text{SAMPLE}$ | 128 | 64 |
| $\rightarrow \text{FC}_{32768} \rightarrow \text{BN} \rightarrow \text{RELU}$ | 32768 | 32768 |
| $\rightarrow \text{RESHAPE}$ | $8 \times 8 \times 512$ | $8 \times 8 \times 512$ |
| $\rightarrow \text{CONVTR}_{256, s2, p1} \rightarrow \text{BN} \rightarrow \text{RELU}$ | $16 \times 16 \times 256$ | $16 \times 16 \times 256$ |
| $\rightarrow \text{CONVTR}_{128, s2, p1} \rightarrow \text{BN} \rightarrow \text{RELU}$ | $32 \times 32 \times 128$ | $32 \times 32 \times 128$ |
| $\rightarrow \text{CONVTR}_{64, s2, p1} \rightarrow \text{BN} \rightarrow \text{RELU}$ | - | $64 \times 64 \times 64$ |
| $\rightarrow \text{CONV}_{64, s1, p1}$ | $32 \times 32 \times 64$ | $64 \times 64 \times 64$ |
| $\rightarrow \text{MLR}$ | $32 \times 32 \times 3$ | $64 \times 64 \times 3$ |

Table 6: Summary of hyperparameters used in training image generation models.

| Hyperparameter | MNIST | CIFAR-10/100 | CELEBA |
|---|---|---|---|
| Epochs | 100 | 100 | 70 |
| Batch size | 100 | 100 | 100 |
| Learning rate | 5e-4 | 5e-4 | 5e-4 |
| Weight decay | 0 | 0 | 0 |
| KL loss weight | 0.312 | 0.024 | 0.09 |
| Optimizer | (Riem)Adam | (Riem)Adam | (Riem)Adam |
| Floating point precision | 32 bit | 32 bit | 32 bit |
| GPU type | RTX A5000 | RTX A5000 | RTX A5000 |
| Num. GPUs | 1 | 1 | 2 |

# D  ABLATION EXPERIMENTS

## D.1  HYPERBOLICITY

Motivating the use of HNNs by measuring the $\delta$-hyperbolicity of visual datasets was proposed by Khrulkov et al. (2020). The idea is to first generate image embeddings from a vision model and then quantify the degree of inherent tree-structure. This is achieved by considering $\delta$-slim triangles and determining the minimal value that satisfies the triangle inequality using the Gromov product. The lower $\delta \geq 0$ is, the higher the hyperbolicity of the dataset. Usually, the scale-invariant value $\delta_{rel}$ is reported. For further insights refer to Khrulkov et al. (2020).

While Khrulkov et al. (2020) studies the hyperbolicity of the final embeddings, we extend the perspective to intermediate embeddings from the encoder, aiming to provide additional motivation for fully hyperbolic models. For this, we first train a Euclidean ResNet-18 classifier using the settings detailed in Appendix C.1. Then, we run the model on the test dataset and extract the embeddings after each ResNet block to assess their $\delta$-hyperbolicity. The results in Table 7 show that all intermediate embeddings exhibit a high degree of hyperbolicity. Furthermore, we observe a difference in hyperbolicity between blocks. This motivates hybrid hyperbolic encoders (HECNNs) with hyperbolic layer only used where hyperbolicity is the highest.

Table 7: $\delta_{rel}$ for intermediate embeddings of ResNet-18 trained on CIFAR-100.

| Encoder Section | $\delta_{rel}$ |
|---|---|
| Initial Conv. | 0.26 |
| Block 1 | 0.19 |
| Block 2 | 0.23 |
| Block 3 | 0.18 |
| Block 4 | 0.21 |

## D.2  HCNN COMPONENTS

In this section, we perform ablation studies to obtain additional insights into our proposed HCNN components. All ablation experiments consider image classification on CIFAR-100 using ResNet-18. We estimate the mean and standard deviation from five runs, and the best performance is highlighted in bold.

**Runtime**  Currently, two major drawbacks of HNNs are relatively high runtime and memory requirements. This is partly due to custom Pythonic implementations of hyperbolic network components introducing significant computational overhead. To study the overhead in practice and assess the efficiency of our implementations, we use PyTorch's *compile* function, which automatically builds a more efficient computation graph. We compare the runtime of our Lorentz ResNets with the Euclidean baseline under compiled and default settings in Table 8 using an RTX 4090 GPU.

Table 8: Runtime improvement using PyTorch's *compile* function on ResNet-18. Duration of a training epoch in seconds.

| | Euclidean | Hybrid L. | HCNN | HECNN |
|---|---|---|---|---|
| Default | **7.4** | 9.2 | 103.3 | 65.2 |
| Compiled | **7.1** | 8.0 | 62.1 | 42.5 |
| Difference | -4.1% | -13.0% | **-39.9%** | -35.8% |

The results show that hybrid HNNs only add little overhead compared to the significantly slower HCNN. This makes scaling HCNNs challenging and requires special attention in future works. However, we also see that hyperbolic models gain much more performance from the automatic compilation than the Euclidean model. This indicates greater room for improvement in terms of implementation optimizations.

**Batch normalization** We investigate the effectiveness of our proposed Lorentz batch normalization (LBN) compared to other possible methods. Specifically, we train HCNN-ResNets with different normalization methods and compare them against a model without normalization. The results are shown in Table 9.

Firstly, the batch normalization in tangent space and the Riemannian batch normalization (Lou et al., 2020) lead to infinite loss within the first few training iterations. This could be caused by the float32 precision used in this work. Additionally, the normalization within the Lorentz fully-connected layer (LFC) proposed by Chen et al. (2021) (see Section A.2) inhibits learning when not combined with our LBN.

Table 9: Normalization ablation.

| Normalization | Accuracy (%) |
|---|---|
| None | $42.24_{\pm 3.86}$ |
| Space bnorm | $77.48_{\pm 0.25}$ |
| Tangent bnorm | NaN |
| Riemannian bnorm (Lou et al., 2020) | NaN |
| LFC norm (Chen et al., 2021) | $1.00_{\pm 0.00}$ |
| LBN + LFC norm (Chen et al., 2021) | $76.98_{\pm 0.18}$ |
| LBN | $\mathbf{78.07_{\pm 0.21}}$ |

Conversely, using LBN alone improves convergence speed and accuracy significantly, getting approximately 36% higher accuracy than the non-normalized model after 200 epochs. Combining LBN and the LFC normalization leads to worse accuracy and runtime, validating our modified LFC (see Section A.2). Overall, this experiment shows the effectiveness of our LBN and suggests that, currently, there is no viable alternative for HNNs using the Lorentz model. However, a naive implementation operating on the hyperbolic space component can serve as a good initial baseline, although ignoring the properties of hyperbolic geometry.

**Non-linear activation** In this work, we propose a method for applying standard activation functions to points in the Lorentz model that is more stable and efficient than the usual tangent space activations. Here, we quantify the improvement by comparing HCNN-ResNets using different ReLU applications. Furthermore, we consider a model without activation functions, as the LFC is non-linear already and might not need such functions.

Table 10: Activation ablation.

| Activation | Accuracy (%) |
|---|---|
| None | $52.85_{\pm 0.41}$ |
| Tangent ReLU | $77.43_{\pm 0.45}$ |
| Lorentz ReLU | $\mathbf{78.07_{\pm 0.21}}$ |

The results in Table 10 show that non-linear activations improve accuracy significantly and are therefore needed in HCNNs. Furthermore, compared to Tangent ReLU, our Lorentz ReLU increases the average accuracy by 0.64%, decreases the runtime of a training epoch by about 16.7%, and is more consistent overall.

**Residual connection** In this experiment, we compare the effect of different approaches for residual connections on performance. As mentioned in Section 4.4, vector addition is ill-defined in the Lorentz model. As alternatives, we test tangent space addition (Nickel & Kiela, 2018), parallel transport (PT) addition (Chami et al., 2019), Möbius addition (after projecting to the Poincaré ball) (Ganea et al., 2018), fully-connected (FC) layer addition (Chen et al., 2021), and our proposed space component addition. The results are shown in Table 11.

In training, we observe that PT and Möbius addition make the model very unstable, causing an early failure in the training process. This might be because of the float32 precision again. The tangent space addition performs relatively well, but the needed exponential and logarithmic maps add computational overhead ($\approx$ 12% higher runtime per training epoch) and some instability that hampers learning. The FC addition and our proposed space addition are very similar and perform best. However, our method is simpler and therefore preferred.

Table 11: Residual connection ablation.

| Residual connection | Accuracy (%) |
|---|---|
| Tangent addition (Nickel & Kiela, 2018) | $77.73_{\pm 0.32}$ |
| PT addition (Chami et al., 2019) | NaN |
| Möbius addition (Ganea et al., 2018) | NaN |
| FC addition (Chen et al., 2021) | $77.93_{\pm 0.25}$ |
| Space addition | $\mathbf{78.07_{\pm 0.21}}$ |

**Initialization** Chen et al. (2021) proposed initializing Lorentz fully-connected layers with the uniform distribution $\mathcal{U}(-0.02, 0.02)$. As the LFC is the backbone of our Lorentz convolutional layer, we test the uniform initialization in HCNNs and compare it to the standard Kaiming initialization (He et al., 2015a) used in most Euclidean CNNs. For this, we employ the same ResNet architecture and initialize Lorentz convolutional layers with these two methods.

The results in Table 12 show that the Kaiming initialization is preferred in HCNNs, leading to 0.55% higher accuracy.

Table 12: Initialization ablation.

| Initialization | Accuracy (%) |
|---|---|
| $\mathcal{U}(-0.02, 0.02)$ (Chen et al., 2021) | $77.52_{\pm 0.10}$ |
| Kaiming (He et al., 2015a) | $\mathbf{78.07_{\pm 0.21}}$ |

