# OpenReview forum: "Fully Hyperbolic Convolutional Neural Networks for Computer Vision"
_ICLR.cc/2024/Conference — ICLR 2024 poster_

### Official Review · Reviewer_QXC6 · 2023-10-29

**Soundness:** 2 fair
**Presentation:** 2 fair
**Contribution:** 3 good
**Rating:** 6
**Confidence:** 4

**Summary:**

This work proposes a fully hyperbolic convolutional network based on the Lorentz model of hyperbolic geometry. Lorentz formulations of Euclidean operations and extensions from the Poincare reformulations of operations such as MLR, FC, and Concat are presented. A series of networks are presented and evaluated in classification tasks, under adversarial attacks and reconstruction tasks. \

Contributions:
Fully Hyperbolic convolutional neural network outperforming Poincare alternatives. \

Reformulation of core operations of convolutional neural networks for the Lorentz model. \

Lorentz batch normalization is proposed based on centroid. \

Lorentz residual and activation functions are proposed based on space-time vector decomposition. \

**Strengths:**

Many of the reformulations are sensible and follow well-established reformulations of hyperbolic deep learning that are widely applicable in hyperbolic deep learning in Lorentz models. \

A fully Lorentz formulation of convolutional networks is missing in the hyperbolic deep learning field and is likely to be an essential component given the improved stability of such models of hyperbolic space. Hence, this work presents a good addition to the literature. \

Using Lorentzian centroid in the normalization is sensible and alleviates many of the computation restrictions of the frechet mean on the Poincare model. This normalization scheme seems reasonable and computationally efficient (more empirical studies to show performance would be a nice addition). \

The experimentation demonstrates good performance increases under fully Lorentz models, although this could be expanded. \

The work for the most part is well written and nicely organized. \

**Weaknesses:**

The claims of novelty regarding the first fully hyperbolic and first hybrid CNN for vision tasks do not hold true, you do not cite, nor address the work [1]. Additionally, you reference and discuss a survey of hyperbolic deep learning for vision which contains a significant number of hybrid architectures for vision [2]. \

The same issue holds true for batch-normalization regarding missing literature, where [1] also presented a method that does not rely on frechet mean, however, your method is the first to my knowledge to operate in the Lorentz model via such a method. \

The experimental ablations and sensitivity are extremely limited, for example, I would expect at the least to see experiments analyzing their performance with and with the batch norm, as there exist many works that suggest batch norm is not necessary [2]. In addition, further analysis into each component, and architectural setting would be a useful insight for those wanting to utilize this work, i.e. initialisations, and curvatures. \

No comparison is made to existing hyperbolic CNN operations i.e. Hyperbolic neural networks ++, and Poincare Resnet. Although not operating in the same hyperbolic model which is notably more unstable, a comparison would help demonstrate the benefits of your proposal. \

You mention that alternative residual connection methods perform worse, however no results are shown for this analysis, these missing experimentation (even if placed in the supplementary) would provide a more compelling analysis. In addition, a comparison to the Poincare midpoint [1] could be made given the Lorentz model is isometric to Poincare. \

The choice of FID is a debated metric, it has been shown that this metric does not appropriately represent the quality of generations [3]. It would be nice to see a series of the generations to compare human perception of the generations. \

You refer to the benefit of capturing hierarchies, yet your empirical studies do not support /compare the capturing of hierarchies of any kind to support this claim. \

[1] van Spengler, M., Berkhout, E. and Mettes, P., 2023. Poincar\'e ResNet. arXiv preprint arXiv:2303.14027.

[2] Shimizu, R., Mukuta, Y. and Harada, T., 2020. Hyperbolic neural networks++. arXiv preprint arXiv:2006.08210.

[3] Jung, S. and Keuper, M., 2021, December. Internalized biases in fréchet inception distance. In NeurIPS 2021 Workshop on Distribution Shifts: Connecting Methods and Applications.

**Questions:**

Do you observe any need for hyperbolic non-linear activation functions given that hyperbolic networks are inherently non-linear? Some works in Poincare ball suggest that it's not essential but marginally improves performance. \

You interchangeably reference Hybrid Lorentz as your own work and also reference (Nagano et al., 2019), can you define how yours differs? \

In the re-scaling procedure, you assume that the variance direction is along the geodesic intersecting the origin, however, this may not be the case, therefore is not an accurate formulation. Can you elaborate if I have mis-understood. \

Can you expand on the residual connection and activation section, currently it is unclear how the time and space components (dimensions of the point) can be added without the issues that arise in hyperbolic space? You suggest that the space component lies in Euclidean space, however simply decomposing the hyperbolic vector based of the first dimension to achieve this space and time component does not align to conventional works. I would be grateful if you could elaborate. \

---

> ### Author Response · Authors · 2023-11-16
>
> Thank you very much for going through the paper and giving an in-depth review. The comments you make are very on point and show consideration for the paper.
>
> ## Weaknesses
>
> > 1. & 2. The claims of novelty regarding the first fully hyperbolic, first hybrid CNN for vision tasks and batch normalization do not hold true, because of Poincaré ResNet [1] and other hybrid architectures for vision.
>
> We actually published a preprint of our work in March of this year, within days of the Poincaré ResNet paper, making the “being first” claim debatable. We think it is a concurrent work, but we will definitely add it to the related works. However, our novelty claims always hold for the Lorentz model, i.e., we proposed the first fully hyperbolic Lorentz CNN, Lorentz convolutional layer, Lorentz batch normalization layer, Lorentz MLR, etc. In the revised version, we tried to weaken our claim of being the first a bit.
>
> Nevertheless, we agree that a comparison to the Poincaré ResNet would be very interesting. With the official code now available, we adjusted it for the classification task and ran some quick experiments. For the Poincaré ResNet, we observed a much slower compute time, double memory requirements, and 94.5% and 76.6% accuracy for CIFAR-10 and CIFAR-100, respectively, which is below the Euclidean and the hybrid Poincaré models. We have started integrating these empirical results into the current version of the paper and will continue to experiment for Tiny-ImageNet, adversarial robustness, low embedding dimensions, and runtime comparisons.
>
> On the other side, we do not claim to propose the first hybrid hyperbolic CNN, in general (as evident from the related works section). We claim to propose the first hybrid CNN with a hyperbolic encoder. This may be in competition with the concurrent Poincaré ResNet work, but not with any other work.
>
> > 3. & 5. Experimental ablations are limited. Further analysis into each component and architectural setting would be a useful insight for those wanting to utilize this work, i.e. batch normalization, initializations, and curvatures. No results for alternative residual connection.
>
> Ablation experiments with respect to individual components and architecture settings are indeed important. We had already included them in the appendix. Architectural settings are given in Appendix C, and we conducted many ablation experiments in Appendix D. We tested many different architectural settings, i.e., batch normalizations, activation functions, residual connections, and initializations.
>
> Regarding curvature, we mentioned in Appendix C that Lorentz HNNs perform best with $K=-1$ and Poincaré HNNs with $K=-0.1$.
>
> > 4. & 5. Similar to 1. & 2.: No comparison is made to existing hyperbolic CNN operations, i.e., Hyperbolic neural networks++. Although not operating in the same hyperbolic model which is notably more unstable, a comparison would help demonstrate the benefits of your proposal. In addition, a comparison to the Poincare midpoint [1] could be made given the Lorentz model is isometric to Poincare.
>
> The hyperbolic neural networks++ paper (Shimizu et al., 2020) proposed 1D convolutions for the Poincaré ball and NLP. To compare against their method, we would have needed to design the other CNN components in the Poincaré ball (Poincaré batch norm, Poincaré residuals, etc.).
>
> The other option would be a hybrid Lorentz-Poincaré network by using the Poincaré convolution of Shimizu et al. in our model. However, this is not feasible because, while the hyperbolic models are isometric, such a network would require many mapping operations between the models, which is very bad for runtime, numerical precision, and stability.
>
> Similarly, computing the Poincaré midpoint for points in the Lorentz model would require additional mapping operations. Furthermore, the Poincaré midpoint and the Lorentzian centroid are equivalent and exactly match each other (Peng et al., 2021). So, this method would only add additional operations while computing the exact same point.
>
> As mentioned before, we are including a comparison to Poincaré ResNet.
>
> > 6. The choice of FID is a debated metric, it would be nice to see a series of the generations to compare human perception of the generations.
>
> We agree. However, considering the small network scale at the moment, the images generated hold little insight. It is difficult to relate output quality to differences in model architecture (Euclidean versus hyperbolic) for such small model sizes.

---

> ### Author Response · Authors · 2023-11-16
>
> > 7. Your empirical studies do not support /compare the capturing of hierarchies of any kind.
>
> We acknowledge that there are no explicit hierarchy experiments. However, most hierarchies in images are intrinsic (on a localization level). To reflect this, we added the measured hyperbolicity of each dataset, the lower the $\delta_{rel}>0$ the higher the hyperbolicity/tree-like structure of image features (see Table 2 and Appendix D.1). Additionally, all datasets include hierarchical relations between object classes (e.g., superclass label in CIFAR-100 (https://www.cs.toronto.edu/~kriz/cifar.html)). Only CIFAR-10 might have too few classes for any significant object class relations.
>
> We revised the experiments section and Appendix D.1 to make our points clearer.
>
> ## Questions
>
> > 1. Is hyperbolic non-linear activation needed in HCNN?
>
> We had also wondered how important the activation function was after reading the literature, so we performed some experiments (no activation vs activation on the tangent space, vs direct activations) and have them in Appendix D. We were surprised by the level of effect they actually had on the model as a whole.
>
> > 2. What is the difference between the hybrid Lorentz networks in classification and generation?
>
> In image generation, a hybrid VAE for the Lorentz model already existed. Nagano et al. (2019) defined the wrapped normal distribution in the Lorentz model. Therefore, the hybrid VAE does not contain any of our novel layers.
>
> In classification, only a hybrid Poincaré classifier existed. So, for comparison, we used our Lorentz MLR to obtain a hybrid Lorentz classifier.
>
> > 3. In re-scaling of batch normalization, why do you assume the variance direction is along the geodesic intersecting the origin?
>
> As we measure the distance from the Lorentzian centroid to the points to obtain the variance in batch normalization, the variance geodesics always intersect their centroid. So, if the centroid does not lie at the origin, the variance geodesics do not intersect the origin. That is why, before re-scaling, we move the centroid to the origin by parallel transporting the points. Now, all variance geodesics intersect the origin.
> Maybe the following figure helps: 1. Compute the centroid (variance geodesics do not intersect the origin), 2. Parallel transport points towards the origin (variance geodesics intersect the origin), 3. Use tangent space for rescaling (variance geodesics are straight lines). 4. Parallel transportation to the new centroid (not shown in the figure).
>
> [![unnamed.png](https://i.postimg.cc/76vLGs9s/unnamed.png)](https://postimg.cc/7CV4jNp0)
>
>
> > 4. Regarding residual connection and activation in Section 4.4: Decomposing the hyperbolic vector based on the first dimension to achieve this space and time component does not align with conventional works.
>
> The general idea of this approach was first presented by Chen et al. (2021) in their definition of the Lorentz fully-connected layer. They note that this can be considered a pseudo-hyperbolic operation. We interpret this as similar to a transformation parameterized by the residual value.
>
> In Appendix D, we studied the effect of different hyperbolic residual connections and activation functions, showing the advantages of our definition.

---

> > ### Comment · Reviewer_QXC6 · 2023-11-22
> > **Thank you**
> >
> > I appreciate your responses and the issue concerning the concurrent work.
> > Happy to increase my score

---

### Official Review · Reviewer_T97r · 2023-10-30

**Soundness:** 3 good
**Presentation:** 3 good
**Contribution:** 3 good
**Rating:** 6
**Confidence:** 4

**Summary:**

In this paper, the  authors propose the first fully hyperbolic convolutional neural network encoder in computer vision. In particular, the authors introduce Lorentzian formulations of the 2D convolutional layer, batch normalization, and multinomial logistic regression for constructing the fully hyperbolic convolutional neural network. Extensive experiments show that the proposed method is better than existing hybrid hyperbolic convolutional neural network.

**Strengths:**

1. The paper introduced the Lorentzian formulations of the 2D convolutional layer, batch normalization, and multinomial logistic regression which are new in the literature of hyperbolic neural networks.

2. The authors conducted extensive experiments (image classification, image generation) to demonstrate the effectiveness of the proposed HCNN Lorentz.

**Weaknesses:**

1. Across all the experiments, the improvement of the proposed HCNN Lorentz/HECNN Lorentz over the standard Euclidean neural network is usually small (65.96 vs. 65.19 on Tiny-ImageNet), so it is not clear what is the practical advantage of the proposed method over standard Euclidean neural network.

2. There is a lack of in-depth analysis of the experimental results. For example, the results show that HECNN is the best when the embedding dimension is low. However, the reason of such an improvement is not clear. Similarly for the image generation task.

3. It is known that hyperbolic space is well suited for hierarchical dataset, however, none of the experiments clearly demonstrate this.

**Questions:**

1. The authors show that fully  hyperbolic neural networks is better than the hybrid version, however, would fully  hyperbolic neural networks incur more parameters than the hybrid version? Also, how about the time complexity comparison？

2. It is not clear what is the real difference made by Lorentzian convolutional layer compared with the standard convolutional layer, more analysis and visualizations are needed to understand the real benefits of the proposed approaches.

---

> ### Author Response · Authors · 2023-11-16
>
> We appreciate the thorough review and constructive feedback provided by the reviewer. We are happy to respond to your questions below.
>
> ## Weaknesses
>
> > 1. HCNN gives only small performance improvements.
>
> Citing our response to reviewer Nvg1:
>
> “[...] The improvements of our HECNN compared to the other provided models on CIFAR100 and Tiny-Imagenet are significant. We provide a table with the p-value for the respective unpaired t-tests below:
>
> |                 | CIFAR-100 | Tiny-ImageNet |
> |-----------------|:-----------:|:---------------:|
> | Euclidean       | <0.0001   | <0.0001       |
> | Hybrid Poincaré | 0.0002    | 0.0006        |
> | Hybrid Lorentz  | 0.0009    | 0.0072        |
> | HECNN            | -         | -             |
> | HCNN            | 0.0008    | 0.0359        |
>
> We also provide a table with the models vs Euclidean performance here:
>
> |                 | CIFAR-100 | Tiny-ImageNet |
> |-----------------|:-----------:|:---------------:|
> | Euclidean       | -         | -             |
> | Hybrid Poincaré | 0.053     | 0.1827        |
> | Hybrid Lorentz  | 0.0277    |  0.0002       |
> | HECNN           | <0.0001   | <0.0001       |
> | HCNN            | 0.0087    |  0.0002       |
>
> These indicate very high statistical significance for HECNN vs all other models and high significance for Lorentz models vs Euclidean. This is also important as no HNN from the literature had outperformed Euclidean models on standard classification before.”
>
>  We also try to emphasize the advantageous properties of HNNs. The proposed Lorentz models show a much larger performance gap for adversarial robustness tasks and when using lower dimensional embeddings. One more thing to note is that this work is also geared toward providing the building blocks for further hyperbolic work for vision models. These tools could be used to create specialized approaches later on.
>
> > 2. Lack of in-depth analysis of experimental results.
>
> The lack of extensive motivation and analysis was a result of the page limit. We tried to improve it in the revised manuscript (see, e.g., Section 5.1 and Appendix D.1 for hyperbolicity motivation). We now try to include the motivation, hypothesis, and more analysis for every experiment.
>
> > 3. No experiment clearly demonstrates that hyperbolic space is well-suited for hierarchical datasets.
>
> We acknowledge that there are no explicit hierarchy experiments. However, most hierarchies in images are intrinsic (on a localization level). To reflect this, we added the measured hyperbolicity of each dataset, the lower the $\delta_{rel}>0$ the higher the hyperbolicity/tree-like structure of image features (see Table 2 and Appendix D.1). Additionally, all datasets include hierarchical relations between object classes (e.g., superclass label in CIFAR-100 (https://www.cs.toronto.edu/~kriz/cifar.html)). Only CIFAR-10 might have too few classes for any significant object class relations.
>
> We revised the experiments section to make our points clearer.
>
> ## Questions
>
> > 1. Do HCNNs incur more parameters? Time complexity comparison?
>
> Overall, HCNNs have no major impact on the number of parameters.
>
> However, time complexity is certainly still an issue. The convolutional layer is the main bottleneck both in terms of computation time and memory requirements. This is generally because we lose the heavily optimized closed-source CUDA convolution implementations when we replace PyTorch’s built-in convolution. To verify this, we reimplemented the Euclidean convolution with the same naive operation breakdown (Unfold + matrix multiplication)  and it is around 5.5x slower when compared to the built-in convolution (averaged over 2000 runs with a 3x3 kernel). Thus, most of the computational overhead of our HCNN appears to be implementation-related. We verify this through the use of PyTorch 2.0’s inbuilt “compile” which is able to drop the training time of the hyperbolic mode by around 40% by attempting to find mergeable operations in the computation graph before runtime (as we can see in Appendix D).
> To verify this we also measure the FLOPs of all 3 convolution types (nn.Conv2d, reimplemented conv2d and Lorentz conv2d) as reported by the pytorch profiler. This was for an input of size 128x64x64x3 and kernel size 3x3 with 0 padding. All were approximately around 1.7 GFLOPs (with the lorentz conv being slightly less even) despite the drastic time and memory differences.
>
> > 2. What is the difference between the Lorentz convolutional Layer and the Euclidean convolutional layer?
>
> We try to make the hyperbolic operations as 1 to 1 as possible. As such, the main difference is that in the Lorentz convolutional layer, the features lie in the Lorentz model, not the Euclidean space. The unfolding and transformations are also performed in a manner that preserves the hyperbolic feature space.

---

### Official Review · Reviewer_Nvg1 · 2023-11-03

**Soundness:** 3 good
**Presentation:** 2 fair
**Contribution:** 4 excellent
**Rating:** 6
**Confidence:** 3

**Summary:**

The paper describes layers for neural networks that operate with hyperbolic features, instead of the standard Euclidean vector features. The paper presents Convolution layers, Batch-normalization, activation function, skip connections and formula for hyperbolic decision boundaries and logits. The hyperbolic features are describe via a Lorentz model, which, compared to the Poincare model, had improved numerical stability and convenient closed form expressions for certain operations. The paper then experimentally validates the soundness of their approach in comparison to an Euclidean baseline (same architecture otherwise) and hybrid adaptions that combine Euclidean layers with hyperbolic layers. The results show a modest improvement in performance (though might not be significant), and increase robustness against adversarial attacks. With VAE experiments the authors show improved reconstruction metrics as well as a qualitatively nicer organization of the latent space.

**Strengths:**

* Although the first part of the methodology section makes too much use of references to the appendix, it is generally clearly written and accessible.
* The authors provide answers to a so-far unaddressed problem (will the use of hyperbolic representations be useful beyond just as an embedding space, but instead use throughout the network)
* The paper is self-contained and is accompanied with code

**Weaknesses:**

1. I find the evidence for the actual usefulness of hyperbolic representations not convincing, in contrast to what the authors present in the abstract and introduction by saying things like "demonstrate the superiority of our HCNN framework". This is not to downplay the results, I still think it is valuable that these original approaches are developed and explored, however, I think some conclusions could be nuanced and perhaps more precisely formulate. Which brings me to the next item
2. It is not always clear what the hypotheses are. It seems like it is "just interesting" to try and build a fully hyperbolic network, without clearly specifying why this would be useful. In particular, the experiments should be accompanied with clear hypotheses of what is expected (why would the Lorentz model be better than the Poincare model if both are isometric representations of the same thing? Why would one expect HE or H model be better compared to the other? Why would the H model be more robust to adversarial attacks?). It would help if the experiments are more clearly motivated, and explain to what questions these experiments provide answers
3. The results, in particular in Table 1, all look the same to me. Almost all results lie within each others uncertainty margins and hence drawing conclusions based on these results is dubious, especially if they as strongly state as on some instances in the paper.
4. It would be great if some very narrowly defined experiments could be defined to test for desirable properties of the network (if this is possible). Because, the performances are all so close to each other which I think has to do with the fact that almost any neural network can get great results if you engineer it well enough, even if the code contains bugs or design flaws! However, in order to define such experiments one first has to clearly specify what unique properties are obtained by the HCNNs, which I couldn't really get from the paper, nor did I find concrete research questions that are being addressed.
5. In the Lorentz model section, Figure 4 is reference. It is not nice to refer to a figure which isn't even included in the document. Moreover, the proper way of refering to figures in order of appearance (e.g. first refer to figure 1, then 2 then 3 etc. Not starting with fig 4). I think if this figure is that important it should be included in the main body. Moreover, it would have been nice to include the appendix in the main pdf, not as a separate document.

**Questions:**

1. The contributions state "superior performance" without specifying relative to what? Regardless of the comparison, this statement should be nuanced given the discussion above.
2. Page 3, on the Lorentz model. What is "the upper sheet" and what is meant with a "two-sheeted hyperboloid"?
3. In Equation 1 it says $x_t > 0$. Is this because $x_t \geq \sqrt{-1/K}$, and if so, why not specify that instead?
4. Section 4.1 Formalization of the convolution layer... It says "an affine transformation between a linearized kernel and a ..." an affine transformation is a transformation of something, not between two things right? One could say the transformation of a vector is parametrized by a kernel (and a bias), or am I misunderstanding something here?
4. On the same note, equation 3: The definition of LFC seems essential to me, why is it hidden in the appendix and not the main body? This layer is the main work-horse of the neural network, I imagine.
5. Next, "the correct local connectivity is established", what does that mean?
6. Equation 5, why are there additional | | (vertical bars). I count three of them on each side. Two for the norm, which makes sense, but then another outside of it. Please remove it unless there is a particular reason for it.
7. Regarding the Multinomial Logistic Regression. Is this common? I thought that usually the outputs are parametrized as multi-class Bernoullis with probabilities obtained by soft-max, not channel-wise exponentials. I.e., typically one considers mutually exclusive classes, no? Does there exists a construction for hyperbolic soft-max?
8. In 4.4 you mention "... provides best empirical performance", that is nice. But what is theoretical lost with the approach of simply adding the spatial parts? Is this mathematically allowed? It feels like a hack that could theoretically break certain properties of the network (not sure about this though). I suppose since addition is not defined you need to define an alternative, and this one seems as good as anything, but until this point the motivation has been primarily the idea of parallel transport and Frechet mean type "additions".
9. In section 5.1 you write "replace only the resnet encoder blocks with the highest hyperbolicity", what is "hyperbolicity? What is $\delta_{rel}$, and why would different blocks have different hyperbolicity? Many details are missing here.
10. Also, when working with hybrid models, how does one convert a hyperbolic representation to an Euclidean one?
11. Main result section: "This suggests that the Lorentz model is better suited for HNNs than the Poincaré model", was this expected, why or why not?
12. It says the Hybrid Poincaré model is inferior to the Euclidean case. But isn't this also the case for the Hybrid Loretnz model on Cifar-10, and for the other tasks the difference seems not significant. As stated in the limitations sections above, I think the results could be more critically assessed and conclusions should be nuanced at certain locations. I might of course be mistaken in my analyses, in which case it means the paper could improve still a bit in terms of presentation and discussion of the results :)
13. Regarding robustness experiments. Why is this a relevant experiment?
14. Just above the conclusion "As these structures cannot be found for the hybrid model, ... little impact ..." I do not understand this sentence, could this be clarified? Thank you!

---

> ### Author Response · Authors · 2023-11-16
>
> Thank you very much for your great suggestions. We integrated many of them into our revised version! We’re pretty enthusiastic about the subject, so we’re more than happy to try to answer all of your questions.
>
> ## Weaknesses
>
> > 1. - 4. The main results are very close and do not show clear “superiority” (see CIFAR-10). So, some conclusions could be nuanced and perhaps more precisely formulated. Additionally, almost any neural network can get great results if you engineer it well enough.
>
> We agree that our claims and conclusions should be more nuanced and tried to implement your suggestions into the revised version.
>
> In terms of classification results in Table 1,
> - CIFAR-10 is presented as more of a toy example. Accuracy on CIFAR-10 is nearly saturated, and it becomes difficult to get improvements with the same model size.
> - The improvements of our HECNN compared to the other provided models on CIFAR100 and Tiny-Imagenet are significant. We provide a table with the p-value for the respective unpaired t-tests below:
>
> |                 | CIFAR-100 | Tiny-ImageNet |
> |-----------------|:-----------:|:---------------:|
> | Euclidean       | <0.0001   | <0.0001       |
> | Hybrid Poincaré | 0.0002    | 0.0006        |
> | Hybrid Lorentz  | 0.0009    | 0.0072        |
> | HECNN            | -         | -             |
> | HCNN            | 0.0008    | 0.0359        |
>
> We also provide a table with the models vs Euclidean performance here:
>
> |                 | CIFAR-100 | Tiny-ImageNet |
> |-----------------|:-----------:|:---------------:|
> | Euclidean       | -         | -             |
> | Hybrid Poincaré | 0.053     | 0.1827        |
> | Hybrid Lorentz  | 0.0277    |  0.0002       |
> | HECNN           | <0.0001   | <0.0001       |
> | HCNN            | 0.0087    |  0.0002       |
>
> These indicate very high statistical significance for HECNN vs all other models and high significance for Lorentz models vs Euclidean. This is also important as no HNN from the literature had outperformed Euclidean models on standard classification before.
>
>  One more thing to note is that this work is also geared toward providing the building blocks for further hyperbolic work for vision models. These tools could be used to create specialized approaches later on.
>
> Concerning over-engineering, we aimed to avoid this by using the best known hyperparameters of the Euclidean model during the training of our hyperbolic models. We also make the model architectures as one-to-one as possible.
>
> > 2. & 4. The hypothesis why to build a fully hyperbolic model is not clear. Experiments should be more clearly motivated. Some very narrowly defined experiments could be defined to test for desirable properties of the network (if this is possible).
>
> The lack of extensive motivation was a result of the page limit. We tried to improve it in the revised manuscript (see, e.g., Section 5.1 and Appendix D.1 for hyperbolicity motivation). We now try to include the motivation, hypothesis, and more analysis for every experiment.
>
> Known desirable properties of HNNs are robustness to adversarial attacks and effectiveness at lower feature dimensionalities. Our experiments (Table 2, Figure 3) are designed to test these and now word them better.
>
> > 5. In the Lorentz model section, Figure 4 is referenced but not in the main document. Appendix should be in the main document.
>
> We agree with you and moved the figure to Section 3. We also moved the Appendix to the main document.
>
> ## Questions
>
> > 1. Similar to weaknesses 1. - 4.
>
> Please see our response to weaknesses 1. - 4.
>
> > 2. & 3. What is "the upper sheet" and what is meant with a "two-sheeted hyperboloid"? In Equation 1 it says $x_t>0$, why not $x_t \geq \sqrt{-1/K}$?
>
> As an example consider the 3D two-sheeted hyperboloid with $K=-1$. The hyperboloid is then the set of points that satisfies the equation $x^2+y^2-z^2=-1$. This gives us the shape we see here: https://nmd.web.illinois.edu/quadrics/hyper2.html.
>
> When we say upper sheet, we mean the set of points that lie above the $z=0$ plane in this case, this is also why we write $x_t>0$ in Eq. (1). While it is correct to write $x_t \geq \sqrt{-1/K}$, using $x_t>0$ seems to be the conventional representation of the upper sheet model in the literature.
>
> > 4. Is saying  "...an affine transformation between a linearized kernel and a ..." correct?
>
> Yeah, you are right. We can refer to a transformation as “being between” the original vector and the resultant vector, but in this context, you are definitely correct. We’ve changed it in the revised edition to “We define the convolutional layer as a matrix multiplication between a linearized kernel and a concatenation of the values in its receptive field, following \citet{shimizu-et-al-2020}.”.

---

> ### Author Response · Authors · 2023-11-16
>
> > 5. Why is the definition of the LFC hidden in the Appendix?
>
> We had a few issues conforming with the page limit on the paper and since the LFC closely follows the definition by Chen et al. (2021), we chose to add it to the appendix. We could add it back into the main paper but we would have to sacrifice other content in its stead. However, we have moved the Appendix to the main document to make it more visible.
>
> > 6. What does  "the correct local connectivity is established" mean?
>
> This simply means that the features are multiplied with the correct kernel parameters (considering, e.g., stride and padding). We used the word “connectivity” because convolutional layers are usually visualized by a sliding kernel connecting to the feature map.
>
> > 7. Eq. 5: Why are there additional | | (vertical bars)?
>
> Eq. 5 defines the Lorentzian centroid and was introduced in Eq. 11 of Law et al. (2019). They note “[...] where $| \lVert a \rVert_{\mathcal{L}} | = \sqrt{| \lVert a \rVert^2_\mathcal{L} | }$ is the modulus of the imaginary Lorentzian norm of the positive time-like vector a.” So the additional vertical bars are more of a mathematical technicality.
>
> > 8. Is Multinomial Logistic Regression common? Why not a hyperbolic softmax?
>
> Multinomial Logistic Regression (MLR) is just another term for Softmax-Regression. So, we actually defined a hyperbolic softmax.
>
> What you mean with “[...] the outputs are parametrized as multi-class Bernoullis with probabilities obtained by softmax [...]” is the same as MLR. You obtain the channel-wise exponentials of the MLR definition when decomposing the softmax function.
>
> > 9. Regarding residual connection in Section 4.4: What is theoretically lost with the approach of simply adding the spatial parts? Is this mathematically allowed? Until this point, the motivation has been primarily the idea of parallel transport and Frechet mean type "additions".
>
> As you said, one of the issues with the residual connection is the lack of actual addition operation in the hyperbolic space. Initially, we thought of replacing it with parallel transport but found that it was both unstable and expensive (see Appendix D). The idea of space addition is similar to that presented by Chen et al. (2021); it is a pseudo-hyperbolic operation. One can consider it mathematically closer to being a transformation parametrized by the residual value. It would then be able to fall under the boost and rotation operations covered by Chen et al. (2021).
>
> > 10. What is "hyperbolicity”? What is $\delta_{rel}$, and why would different blocks have different hyperbolicity? Many details are missing here.
>
> We agree that details on hyperbolicity are missing. That’s why we added a short summary to Appendix D.1.
>
> Hyperbolicity was the main talking point in the paper of Khrulkov et al. (2020). The authors first produce image embeddings from a vision backbone and then measure the “distance distortion” between them. This is done by presenting the data in $\delta$-slim triangle and finding the smallest $\delta$ that validates the triangle inequality postulate using the Gromov product. $\delta_{rel}$ is basically the calculated scale invariant value of $\delta$. This is a very brief explanation, and we refer to the original authors for a much more in-depth analysis.
>
> Khrulkov et al. do this measurement only for the embeddings of the final block. We repeat this measurement on the embedding sets outputted by each of the other blocks as well. These different blocks could have different hyperbolicities because of the underlying features learned by the encoder, the data dimensionality, and model shallowness, which could, in turn, affect the representative capacity of the embeddings, etc…

---

> ### Author Response · Authors · 2023-11-16
>
> > 11. How does one convert a hyperbolic representation to an Euclidean one?
>
> There are two main ways to extract Euclidean values from a hyperbolic network. The first is projecting the values onto the tangent space. Since the tangent space is the Euclidean approximation of the surface, the projected values are entirely Euclidean. The second option is to use distance measures calculated through hyperbolic methods as these are then only values and not “hyperbolic.” Based on prior works (hybrid VAEs), we used the first option.
>
> > 12. Main result section: "This suggests that the Lorentz model is better suited for HNNs than the Poincaré model", was this expected, why or why not?
>
> Prior works showed that the Lorentz model is more numerically stable than the Poincaré ball. We hypothesized this would reduce the inaccuracies and rounding errors that could lead to a degradation in performance.
>
> Numerical stability is an important issue in HNNs. A comprehensive study conducted by Mishne et al. (2022) reveals the following key insights:
>
> 1. The main issue lies in the exponential growth of the hyperbolic space’s volume w.r.t. the radius, and that embeddings are usually pushed towards infinity/the boundary. This introduces numerical instability and rounding errors, particularly when encountering unrepresentable values in floating-point arithmetic (intermediate results and numbers can only be represented with fixed precision, e.g., 32-bit).
> 2. Using the Poincaré ball leads to a “hard” constraint on the representation capacity, where points with a certain distance from the origin will collapse to the boundary. In contrast, using the Lorentz model leads to a “soft” constraint, where some operations can still be performed correctly.
> 3. The Poincaré ball is more susceptible to gradient vanishing. In the Poincaré model, we are often dealing with significantly smaller magnitudes of numbers compared to the Lorentz model. This leads to a bigger gradient vanishing problem, particularly when points are positioned close to the boundary and rounding errors accumulate.
>
> > 13. Similar to weakness (1. - 4.).
>
> See our response to weakness (1. - 4.).
>
> > 14. Regarding robustness experiments. Why is this a relevant experiment?
>
> In the literature, two prominent advantages of using hyperbolic models are robustness to attacks and much better performance given lower-dimensional feature maps within the model. That is why we perform both of these experiments. It helps verify these findings and also further highlights these desirable properties.
>
> We included the motivation in the revised manuscript.
>
> > 15. Just above the conclusion "As these structures cannot be found for the hybrid model, ... little impact ..." I do not understand this sentence, could this be clarified? Thank you!
>
> We meant that the equidistant clusters we found in the latent embedding space of the fully hyperbolic model were not present in the hybrid model. In fact, the shape of the clusters barely changed between the hybrid and the Euclidean model, although the hybrid model learns embeddings in hyperbolic space. This could indicate that including only one hyperbolic layer (as is done in the baselines) is not enough to meaningfully change the internal feature embedding structure of the model. Therefore, its behavior is not really “hyperbolic”.

---

### Author Response · Authors · 2023-11-16
**General Comment**

We thank all reviewers for their insightful comments and constructive feedback, which certainly improved the quality of our paper. We respond to each reviewer separately. We also uploaded a revised version of the manuscript. Please note that, as suggested by the reviewers, we moved the Appendix to the main document for better exposure.

Before each response, we tried to summarize the addressed weakness/question. Please do not hesitate to point out if we did not summarize your point correctly.

---

### Meta-Review · Area_Chair_bdcG · 2023-12-11

**Metareview:**

In this work the authors present a new method for constructing neural networks using new distance metrics for featurizing an image. In particular, instead of a typical Euclidean distance, the work proposes employing hyperbolic features which naturally contain a hierarchical organization. This paper applies this approach to a convolutional neural network and presents hyperbolic variants of convolutions, batch-normalization, activation functions, and classification boundaries. The authors demonstrate their methods on CIFAR-10, CIFAR-100 and Tiny-ImageNet achieving competitive performance as Euclidean-based methods. Additionally, the authors show minor performance gains in terms of robustness to adversarial attacks and improved reconstruction of using generative models.

The reviewers commented positively on the clarity of presentation, the selection of problems that authors consider, the new formulations introduced by the paper, and the extensive experiments. The reviewers also commented negatively on how the authors failed to demonstrate a strong benefit for the hyperbolic geometry over baseline methods. Additionally, the reviewers pointed out how hyperbolic methods typically fare better on tasks in which there exists a hierarchical organization to the problem, which is currently lacking in many of the image classification problems the authors examined. Finally, one reviewer pointed out prior work from another group that may have preceded this work.

Upon my reading of the paper, I agree with all of the reviewer concerns raised about the lack of strong gains and the lack of testing out their method on a problem that may exploit the hierarchical organization of a dataset. That said, my concerns are tempered by the fact that the authors introduce a novel and interesting way of approaching neural networks that may generalize far beyond the image classification problem.

My remaining and primary concern is the issue of prior work surfaced by the one reviewer. The paper [Poincare ResNet](https://openaccess.thecvf.com/content/ICCV2023/html/van_Spengler_Poincare_ResNet_ICCV_2023_paper.html) contains much overlapping material and introduces many similar results based on hyperbolic variants of CNN functions. The paper in question was accepted at ICCV 2023 on July 13 and the main conference occurred on October 4. After checking with the Senior AC and the Program Chairs, I learned that the the official policy is:
> We consider papers contemporaneous if they are published (available in online proceedings) within the last four months. That means, since our full paper deadline is September 28, if a paper was published (i.e., at a peer-reviewed venue) on or after May 28, 2023, authors are not required to compare their own work to that paper.

Hence, the paper presented at ICCV is considered concurrent work. Given that this last concern is remedied, this paper will be accepted at this conference.

**Justification For Why Not Higher Score:**

Results are not terribly strong.

**Justification For Why Not Lower Score:**

Interesting new technique for parameterizing and learning with neural networks.

---

### Decision · Program_Chairs · 2024-01-16

Accept (poster)